# Unpaired Multi-Domain Causal Representation Learning

**Nils Sturma**
Technical University of Munich
Munich Center for Machine Learning
`nils.sturma@tum.de`

**Chandler Squires**
LIDS, Massachusetts Institute of Technology
Broad Institute of MIT and Harvard
`csquires@mit.edu`

**Mathias Drton**
Technical University of Munich
Munich Center for Machine Learning
`mathias.drton@tum.de`

**Caroline Uhler**
LIDS, Massachusetts Institute of Technology
Broad Institute of MIT and Harvard
`cuhler@mit.edu`

## Abstract

The goal of causal representation learning is to find a representation of data that consists of causally related latent variables. We consider a setup where one has access to data from multiple domains that potentially share a causal representation. Crucially, observations in different domains are assumed to be unpaired, that is, we only observe the marginal distribution in each domain but not their joint distribution. In this paper, we give sufficient conditions for identifiability of the joint distribution and the shared causal graph in a linear setup. Identifiability holds if we can uniquely recover the joint distribution and the shared causal representation from the marginal distributions in each domain. We transform our results into a practical method to recover the shared latent causal graph.

## 1 Introduction

An important challenge in machine learning is the integration and translation of data across multiple domains (Zhu et al., 2017; Zhuang et al., 2021). Researchers often have access to large amounts of unpaired data from several domains, e.g., images and text. It is then desirable to learn a probabilistic coupling between the observed marginal distributions that captures the relationship between the domains. One approach to tackle this problem is to assume that there is a latent representation that is invariant across the different domains (Bengio et al., 2013; Ericsson et al., 2022). Finding a probabilistic coupling then boils down to learning such a latent representation, that is, learning high-level, latent variables that explain the variation of the data within each domain as well as similarities across domains.

In traditional representation learning, the latent variables are assumed to be statistically independent, see for example the literature on independent component analysis (Hyvärinen and Oja, 2000; Comon and Jutten, 2010; Khemakhem et al., 2020). However, the assumption of independence can be too stringent and a poor match to reality. For example, the presence of clouds and the presence of wet roads in an image may be dependent, since clouds may cause rain which may in turn cause wet roads. Thus, it is natural to seek a *causal representation*, that is, a set of high-level *causal* variables and relations among them (Schölkopf et al., 2021; Yang et al., 2021b). Figure 1 illustrates the setup of multi-domain causal representation learning, where multiple domains provide different views on a shared causal representation.

Our motivation to study multi-domain causal representations comes, in particular, from single-cell data in biology. Given a population of cells, different technologies such as imaging and sequencing

37th Conference on Neural Information Processing Systems (NeurIPS 2023).

provide different views on the population. Crucially, since these technologies destroy the cells, the observations are uncoupled, i.e., a specific cell may either be used for imaging or sequencing but not both. The aim is to integrate the different views on the population to study the underlying causal mechanisms determining the observed features in various domains (Butler et al., 2018; Stuart et al., 2019; Liu et al., 2019; Yang et al., 2021a; Lopez et al., 2022; Gossi et al., 2023; Cao et al., 2022). Unpaired multi-domain data also appears in many applications other than single-cell biology. For example, images of similar objects are captured in different environments (Beery et al., 2018), large biomedical and neuroimaging data sets are collected in different domains (Miller et al., 2016; Essen et al., 2013; Shafto et al., 2014; Wang et al., 2003), or stocks are traded in different markets.

In this paper, we study identifiability of the shared causal representation, that is, its uniqueness in the infinite data limit. Taking on the same perspective as, for example, in Schölkopf et al. (2021) and Squires et al. (2023), we assume that observed data is generated in two steps. First, the latent variables $Z = (Z_i)_{i \in \mathcal{H}}$ are sampled from a distribution $P_Z$, where $P_Z$ is determined by an unknown structural causal model among the latent variables. Then, in each domain $e \in \{1, \ldots, m\}$, the observed vector $X^e \in \mathbb{R}^{d_e}$ is the image of a subset of the latent variables under a domain-specific, injective mixing function $g_e$. That is,

$$X^e = g_e(Z_{S_e}),$$

where $S_e \subseteq \mathcal{H}$ is a subset of indices. A priori, it is unknown whether a latent variable $Z_i$ with $i \in S_e$ is shared across domains or domain-specific. Even the number of latent variables which are shared across domains is unknown. Moreover, we only observe the marginal distribution of each random vector $X^e$, but none of the joint distributions over pairs $X^e, X^f$ for $e \neq f$. Said differently, observations across domains are unpaired. Assuming that the structural causal model among the latent variables as well as the mixing functions are linear, our main contributions are:

1. We lay out conditions under which we can identify the joint distribution of $X^1, \ldots, X^m$.
2. We give additional conditions under which we are able to identify the causal structure among the shared latent variables.

In particular, identifiability of the joint distribution across domains enables data translation. That is, given observation $x$ in domain $e$, translation to domain $f$ can be achieved by computing $\mathbb{E}[X_f | X_e = x]$. Furthermore, identifying the causal structure among the shared latent variables lets us study the effect of interventions on the different domains.

The main challenge in proving rigorous identifiability results for multi-domain data is that we cannot apply existing results for single-domain data in each domain separately. Even if the causal structure of the latent variables in a single domain is identifiable, it remains unclear how to combine multiple causal structures, i.e., in which way latent variables are shared. We circumvent this problem via a two-step approach: First, we extend the identifiability of linear independent component analysis (Comon, 1994; Hyvärinen and Oja, 2000; Eriksson and Koivunen, 2004; Mesters and Zwiernik, 2022) to the multi-domain setup, which allows us to identify the joint distribution and distinguish between shared and domain-specific latent variables. Moreover, we identify an "overall mixing matrix" and, in a second step, exploit sparsity constraints in this matrix to identify the causal structure among the shared latent variables. This leverages recent results on causal discovery under measurement error in single domains that also exploit sparsity (Xie et al., 2020; Chen et al., 2022; Xie et al., 2022; Huang et al., 2022). Although we emphasize that our focus in this paper is on identifiability, our proofs also suggest methods to learn the joint distribution as well as the shared causal graph from finite samples. We provide algorithms for the noisy setting and, moreover, we analyze how the number of domains reduce uncertainty with respect to the learned representation.

The paper is organized as follows. In the next paragraphs we discuss further related work. Section 2 provides a precise definition of the considered setup. In Section 3 we consider identifiability of the joint distribution. Using these results, we study identifiability of the causal graph in Section 4. We conclude with a small simulation study as a proof of concept for the finite sample setting in Section 5. Due to space constraints, the detailed discussion of the finite sample setting is deferred to the Appendix. Moreover, the Appendix contains all proofs, discussions on the necessity of our assumptions, and additional examples and simulation results.

**Multi-domain Integration.** Motivated by technological developments for measuring different modalities at single-cell resolution, several methods have been proposed recently for domain

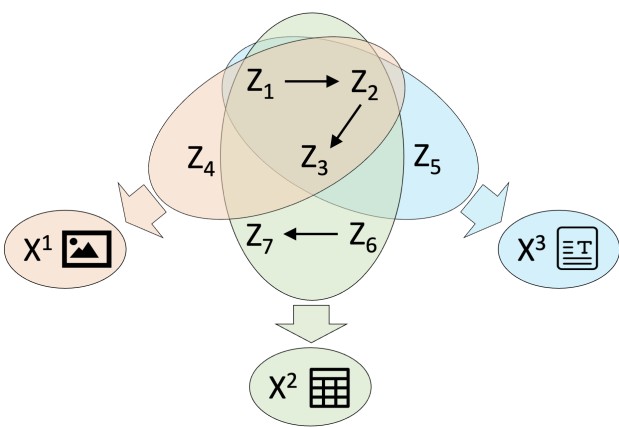

Figure 1: **Setup**. A latent causal representation where multiple domains $X^e$ provide different "views" on subsets of the latent variables $Z_i$. The domains may correspond to different data modalities such as images, text or numerical data. Crucially, the observations across domains are unpaired, i.e., they arise from different states of the latent causal model.

translation between *unpaired* data. The proposed methods rely on a variety of techniques, including manifold alignment (Welch et al., 2017; Amodio and Krishnaswamy, 2018; Liu et al., 2019), matrix factorization (Duren et al., 2018), correlation analysis (Barkas et al., 2019; Stuart et al., 2019), coupled autoencoders (Yang and Uhler, 2019), optimal transport (Cao et al., 2022), regression analysis (Yuan and Duren, 2022), and semisupervised learning (Lin et al., 2022). Implicitly, these methods presume the existence of a *shared* latent space where the different modalities either completely align or at least overlap. However, to the best of our knowledge, none of these methods have rigorous *identifiability* guarantees, i.e., the methods are not guaranteed to recover a correct domain translation mapping even for infinite data. Our work advances the theoretical understanding of multi-domain integration by providing identifiability guarantees on recovering the shared latent space.

**Group Independent Component Analysis.** The primary tool that we use for identifiability is linear independent component analysis (ICA) (Comon, 1994; Eriksson and Koivunen, 2004). Many works extend ICA to the multi-domain setting. These methods primarily come from computational neuroscience, where different domains correspond to different subjects or studies. However, to the best of our knowledge, all prior works require pairing between samples. These works can be categorized based on whether the samples are assumed to be voxels (Calhoun et al., 2001; Esposito et al., 2005), time points (Svensén et al., 2002; Varoquaux et al., 2009; Hyvärinen and Ramkumar, 2013), or either (Beckmann and Smith, 2005; Sui et al., 2009). For reviews, see Calhoun et al. (2009) and Chabriel et al. (2014). Related are methods for *independent vector analysis* (Kim et al., 2006; Anderson et al., 2014; Bhinge et al., 2019) and multiset canonical correlation analysis (Nielsen, 2002; Li et al., 2011; Klami et al., 2014), which allow the latent variables to take on different values in each domain but still require sample pairing. Most of the mentioned methods lack identifiability guarantees, only newer work (Richard et al., 2021) provides sufficient conditions for identifiability. Furthermore, all mentioned methods assume that every latent variable is shared across all domains, while our setup allows for shared and domain-specific latent variables. Some methods, e.g., Lukic et al. (2002), Maneshi et al. (2016), and Pandeva and Forré (2023), permit both shared and domain-specific components, but only consider the paired setting. In this paper, we extend these results to the *unpaired* setting.

**Latent Structure Discovery.** Learning causal structure between latent variables has a long history, e.g., in measurement models (Silva et al., 2006). One recent line of work studies the problem under the assumption of access to interventional data (e.g., Liu et al., 2022; Squires et al., 2023). In particular, Squires et al. (2023) show that the latent graph is identifiable if the interventions are sufficiently diverse. Another line of work, closer to ours and not based on interventional data, shows that the graph is identified under certain sparsity assumptions on the mixing functions (Xie et al., 2020; Chen et al., 2022; Xie et al., 2022; Huang et al., 2022). However, these methods are not suitable in our setup since they require paired data in a single domain. One cannot apply them in each domain separately since it would be unclear how to combine the multiple latent causal graphs, that is, which of the latent variables are shared. In this work, we lay out sparsity assumptions on the mixing

functions that are tailored to the *unpaired multi-domain* setup. The works of Adams et al. (2021) and Zeng et al. (2021) may be considered closest to ours as they also treat a setting with multiple domains and unpaired data. However, our setup and results are more general. Adams et al. (2021) assume that the number of observed variables are the same in each domain, whereas we consider domains of *different dimensions* corresponding to the fact that observations may be of very different nature. Further, we allow for *shared and domain-specific* latent variables, where the number of shared latent variables is unknown, while in Adams et al. (2021) it is assumed that all latent variables are shared. Compared to Zeng et al. (2021), we consider a general but fixed number of observed variables, while Zeng et al. (2021) only show identifiability of the full model in a setup where the number of observed variables in each domain increases to infinity. On a more technical level, the conditions in Zeng et al. (2021) require two pure children to identify the shared latent graph, while we prove identifiability under the weaker assumption of two partial pure children; see Section 4 for precise definitions.

**Notation.** Let $\mathbb{N}$ be the set of nonnegative integers. For positive $n \in \mathbb{N}$, we define $[n] = \{1, \ldots, n\}$. For a matrix $M \in \mathbb{R}^{a \times b}$, we denote by $M_{A,B}$ the submatrix containing the rows indexed by $A \subseteq [a]$ and the columns indexed by $B \subseteq [b]$. Moreover, we write $M_B$ for the submatrix containing all rows but only the subset of columns indexed by $B$. Similarly, for a tuple $x = (x_1, \ldots, x_b)$, we denote by $x_B$ the tuple only containing the entries indexed by $B$. A matrix $Q = Q_\sigma \in \mathbb{R}^{p \times p}$ is a *signed permutation matrix* if it can be written as the product of a diagonal matrix $D$ with entries $D_{ii} \in \{\pm 1\}$ and a permutation matrix $\widetilde{Q}_\sigma$ with entries $(\widetilde{Q}_\sigma)_{ij} = \mathbb{1}_{j=\sigma(i)}$, where $\sigma$ is a permutation on $p$ elements. Let $P$ be a $p$-dimensional joint probability measure of a collection of random variables $Y_1, \ldots, Y_p$. Then we denote by $P_i$ the marginal probability measure such that $Y_i \sim P_i$. We say that $P$ has *independent marginals* if the random variables $Y_i$ are mutually independent. Moreover, we denote by $M \# P$ the $d$-dimensional *push-forward measure* under the linear map defined by the matrix $M \in \mathbb{R}^{d \times p}$. If $Q$ is a signed permutation matrix and the probability measure $P$ has independent marginals, then $Q \# P$ also has independent marginals. For univariate probability measures we use the shorthand $(-1) \# P = -P$.

## 2 Setup

Let $\mathcal{H} = [h]$ for $h \geq 1$, and let $Z = (Z_1, \ldots, Z_h)$ be latent random variables that follow a linear structural equation model. That is, the variables are related by a linear equation

$$Z = AZ + \varepsilon, \tag{1}$$

with $h \times h$ parameter matrix $A = (a_{ij})$ and zero-mean, independent random variables $\varepsilon = (\varepsilon_1, \ldots, \varepsilon_h)$ that represent stochastic errors. Assume that we have observed random vectors $X^e \in \mathbb{R}^{d_e}$ in multiple domains of interest $e \in [m]$, where the dimension $d_e$ may vary across domains. Each random vector is the image under a linear function of a subset of the latent variables. In particular, we assume that there is a subset $\mathcal{L} \subseteq \mathcal{H}$ representing the shared latent space such that each $X^e$ is generated via the mechanism

$$X^e = G^e \cdot \begin{pmatrix} Z_\mathcal{L} \\ Z_{I_e} \end{pmatrix}, \tag{2}$$

where $I_e \subseteq \mathcal{H} \setminus \mathcal{L}$. We say that the latent variable $Z_{I_e}$ are *domain-specific* for domain $e \in [m]$ while the latent variables $Z_\mathcal{L}$ are *shared* across all domains. As already noted, we are motivated by settings where the shared latent variables $Z_\mathcal{L}$ capture the key causal relations and the different domains are able to give us combined information about these relations. Likewise, we may think about the domain-specific latent variables $Z_{I_e}$ as "noise" in each domain, independent of the shared latent variables. Specific models are now derived from (1)-(2) by assuming specific (but unknown) sparsity patterns in $A$ and $G^e$. Each model is given by a "large" directed acyclic graph (DAG) that encodes the multi-domain setup. To formalize this, we introduce pairwise disjoint index sets $V_1, \ldots, V_m$, where $V_e$ indexes the coordinates of $X^e$, i.e., $X^e = (X_v : v \in V_e)$ and $|V_e| = d_e$. Then $V = V_1 \cup \cdots \cup V_m$ indexes all observed random variables. We define an $m$-domain graph such that the latent nodes are the only parents of observed nodes and there are no edges between shared and domain-specific latent nodes.

**Definition 2.1.** Let $\mathcal{G}$ be a DAG whose node set is the disjoint union $\mathcal{H} \cup V = \mathcal{H} \cup V_1 \cup \cdots \cup V_m$. Let $D$ be the edge set of $\mathcal{G}$. Then $\mathcal{G}$ is an $m$-*domain graph* with *shared latent nodes* $\mathcal{L} = [\ell] \subseteq \mathcal{H}$ if the following is satisfied:

1. All parent sets contain only latent variables, i.e., $\mathrm{pa}(v) = \{w : w \to v \in D\} \subseteq \mathcal{H}$ for all $v \in \mathcal{H} \cup V$.

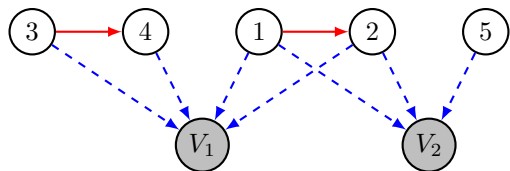

Figure 2: **Compact version of a 2-domain graph** $\mathcal{G}_2$ with five latent nodes and two domains $V_1$ and $V_2$. All observed nodes in each domain are represented by a single grey node. We draw a dashed blue edge from latent node $h \in \mathcal{H}$ to domain $V_e \subseteq V$ if $h \in S_e = \mathrm{pa}(V_e)$. The random vectors associated to the two domains are uncoupled, that is, we do not observe their joint distribution.

2. The set $\mathcal{L}$ consists of the common parents of variables in all different domains, i.e., $u \in \mathcal{L}$ if and only if $u \in \mathrm{pa}(v) \cap \mathrm{pa}(w)$ for $v \in V_e$, $w \in V_f$ with $e \neq f$.

3. Let $I_e = S_e \setminus \mathcal{L}$ be the domain-specific latent nodes, where $S_e := \mathrm{pa}(V_e) = \cup_{v \in V_e}\mathrm{pa}(v) \subseteq \mathcal{H}$. Then there are no edges in $D$ that connect a node in $\mathcal{L}$ and a node $\cup_{e=1}^m I_e$ or that connect a node in $I_e$ and a node in $I_f$ for any $e \neq f$.

To emphasize that a given DAG is an $m$-domain graph we write $\mathcal{G}_m$ instead of $\mathcal{G}$. We also say that $S_e$ is the set of *latent parents* in domain $e$ and denote its cardinality by $s_e = |S_e|$. Note that the third condition in Definition 2.1 does not exclude causal relations between the domain-specific latent variables, that is, there may be edges $v \to w$ for $v, w \in I_e$. Since the sets $I_e$ satisfy $I_e \cap I_f = \emptyset$ for $e \neq f$, we specify w.l.o.g. the indexing convention $I_e = \{\ell + 1 + \sum_{i=1}^{e-1}|I_i|, \ldots, \ell + \sum_{i=1}^{e}|I_i|\}$ and $h = \ell + \sum_{e=1}^{m}|I_e|$.

**Example 2.2.** Consider the compact version of a 2-domain graph in Figure 2. There are $h = 5$ latent nodes where $\mathcal{L} = \{1, 2\}$ are shared and $I_1 = \{3, 4\}$ and $I_2 = \{5\}$ are domain-specific. A full $m$-domain graph is given in Appendix B.

Each $m$-domain graph postulates a statistical model that corresponds to the structural equation model in (1) and the mechanisms in (2), with potentially sparse matrices $A$ and $G^e$. For two sets of nodes $W, Y \subseteq \mathcal{H} \cup V$, we denote by $D_{WY}$ the subset of edges $D_{WY} = \{y \to w \in D : w \in W, y \in Y\}$. Moreover, let $\mathbb{R}^{D_{WY}}$ be the set of real $|W| \times |Y|$ matrices $M = (m_{wy})$ with rows indexed by $W$ and columns indexed by $Y$, such that the support is given by $D_{WY}$, that is, $m_{wy} = 0$ if $y \to w \notin D_{WY}$.

**Definition 2.3.** Let $\mathcal{G}_m = (\mathcal{H} \cup V, D)$ be an $m$-domain graph. Define the map

$$\phi_{\mathcal{G}_m} : \mathbb{R}^{D_{V\mathcal{H}}} \times \mathbb{R}^{D_{\mathcal{H}\mathcal{H}}} \longrightarrow \mathbb{R}^{|V| \times |\mathcal{H}|}$$
$$(G, A) \longmapsto G \cdot (I - A)^{-1}.$$

Then the *multi-domain causal representation (MDCR) model* $\mathcal{M}(\mathcal{G}_m)$ is given by the set of probability measures $P_X = B\#P$, where $B \in \mathrm{Im}(\phi_{\mathcal{G}_m})$ and $P$ is an $h$-dimensional probability measure with independent, mean-zero marginals $P_i, i \in \mathcal{H}$. We say that the pair $(B, P)$ is a *representation* of $P_X \in \mathcal{M}(\mathcal{G}_m)$.

Definition 2.3 corresponds to the model defined in Equations (1) and (2). If $P_X \in \mathcal{M}(\mathcal{G}_m)$ with representation $(B, P)$, then $P_X$ is the joint distribution of the observed domains $X = (X^1, \ldots, X^m)$. The distribution of the random variables $\varepsilon_i$ in Equation (1) is given by the marginals $P_i$. Moreover, for any matrix $G \in \mathbb{R}^{D_{V\mathcal{H}}}$, we denote the submatrix $G^e = G_{V_e, S_e} \in \mathbb{R}^{d_e \times s_e}$ which coincides with the matrix $G^e$ from Equation (2). For the graph in Figure 2, we compute a concrete example of the matrix $B$ in Example B.1 in the Appendix. Importantly, in the rest of the paper we assume to only observe the marginal distribution $P_{X^e}$ in each domain but not the joint distribution $P_X$.

Ultimately, we are interested in recovering the graph $G_{\mathcal{L}} = (\mathcal{L}, D_{\mathcal{L}\mathcal{L}})$ among the shared latent nodes. We proceed by a two-step approach: In Section 3 we recover the representation $(B, P)$ of the joint distribution $P_X$. To be precise, we recover a matrix $\widehat{B}$ that is equal to $B$ up to certain permutations of the columns. Then we use the matrix $\widehat{B}$ to recover the shared latent graph $G_{\mathcal{L}}$ in Section 4 and show that recovery is possible up to trivial relabeling of latent nodes that appear in the same position of the causal order.

## 3 Joint Distribution

To identify the joint distribution $P_X$, we apply identifiability results from linear ICA in each domain separately and match the recovered probability measures $P_i$ for identifying which of them are shared, that is, whether or not $i \in \mathcal{L}$. Let $\mathcal{G}_m$ be an $m$-domain graph with shared latent nodes $\mathcal{L}$, and let $P_X \in \mathcal{M}(\mathcal{G}_m)$ with representation $(B, P)$. Recall that $B = G(I - A)^{-1}$ with $G \in \mathbb{R}^{D_V \mathcal{H}}$ and $A \in \mathbb{R}^{D_{\mathcal{H}} \mathcal{H}}$. We make the following technical assumptions.

(C1) (Different error distributions.) The marginal distributions $P_i, i \in \mathcal{H}$ are non-degenerate, non-symmetric and have unit variance. Moreover, the measures are pairwise different to each other and to the flipped versions, that is, $P_i \neq P_j$ and $P_i \neq -P_j$ for all $i, j \in \mathcal{H}$ with $i \neq j$. Subsequently, we let $d$ be a distance on the set of univariate Borel probability measures such that $d(P_i, P_j) \neq 0$ and $d(P_i, -P_j) \neq 0$ for $i \neq j$.

(C2) (Full rank of mixing.) For each $e \in [m]$, the matrix $G^e = G_{V_e, S_e} \in \mathbb{R}^{d_e \times s_e}$ is of full column rank.

By not allowing symmetric distributions in Condition (C1), we assume in particular that the distributions of the errors are non-Gaussian. Non-Gaussianity together with the assumptions of pairwise different and non-symmetric error distributions allow us to extend the results on identifiability of linear ICA to the unpaired multi-domain setup and to identify the joint distribution. In particular, the assumption of pairwise different error distributions allows for "matching" the distributions across domains to identify the ones corresponding to the shared latent space. Non-symmetry accounts for the sign-indeterminacy of linear ICA when matching the distributions. We discuss the necessity of these assumptions in Remark 3.2 and, in more detail, in Appendix C. Note that Condition (C1) is always satisfied in a generic sense, that is, randomly chosen probability distributions on the real line are pairwise different and non-symmetric with probability one. Finally, Condition (C2) requires in particular that for each shared latent node $k \in \mathcal{L}$ there is at least one node $v \in V_e$ in every domain $e \in [m]$ such that $k \in \text{pa}(v)$.

Under Conditions (C1) and (C2) we are able to derive a sufficient condition for identifiability of the joint distribution. Let $SP(p)$ be the set of $p \times p$ signed permutation matrices. We define the set of signed permutation *block matrices*:

$$\Pi = \{\text{diag}(\Psi_{\mathcal{L}}, \Psi_{I_1}, \ldots, \Psi_{I_m}) : \Psi_{\mathcal{L}} \in SP(\ell) \text{ and } \Psi_{I_e} \in SP(|I_e|)\}.$$

Our main result is the following.

**Theorem 3.1.** *Let $\mathcal{G}_m$ be an $m$-domain graph with shared latent nodes $\mathcal{L} = [\ell]$, and let $P_X \in \mathcal{M}(\mathcal{G}_m)$ with representation $(B, P)$. Suppose that $m \geq 2$ and that Conditions (C1) and (C2) are satisfied. Let $(\widehat{\ell}, \widehat{B}, \widehat{P})$ be the output of Algorithm 1. Then $\widehat{\ell} = \ell$ and*

$$\widehat{B} = B \cdot \Psi \quad \text{and} \quad \widehat{P} = \Psi^{\top} \# P,$$

*for a signed permutation block matrix $\Psi \in \Pi$.*

Theorem 3.1 says that the matrix $B$ is identifiable up to signed block permutations of the columns. Under the assumptions of Theorem 3.1 it holds that $\widehat{B} \# \widehat{P}$ is equal to $P_X$. That is, the joint distribution of the domains is identifiable.

*Remark* 3.2. While Theorem 3.1 is a sufficient condition for identifiability of the joint distribution, we emphasize that pairwise different error distributions are in most cases also necessary; we state the exact necessary condition in Proposition C.1 in the Appendix. Said differently, if one is willing to assume that conceptually different latent variables also follow a different distribution, then identification of these variables is possible, and otherwise (in most cases) not. Apart from pairwise different error distributions, non-symmetry is then required to fully identify the joint distribution whose dependency structure is determined by the shared latent variables. If the additional assumption on non-symmetry is not satisfied, then it is still possible to identify the shared, conceptually different latent variables, which becomes clear by inspecting the proofs of Theorem 3.1 and Proposition C.1. The non-identifiability of the joint distribution would only result in sign indeterminacy, that is, entries of the matrix $\widehat{B}$ could have a flipped sign.

*Remark* 3.3. By checking the proof of Theorem 3.1, the careful reader may notice that the statement of the theorem still holds true when we relax the third condition in the definition of an $m$-domain

---

**Algorithm 1** IdentifyJointDistribution

---

1: **Input:** Probability measures $P_{X^e}$ for all $e \in [m]$.
2: **Output:** Number of shared latent variables $\widehat{\ell}$, matrix $\widehat{B}$ and probability measure $\widehat{P}$ with independent marginals.
3: **for** $e \in [m]$ **do**
4:   Linear ICA: Find the smallest value $\widehat{s}_e$ such that $P_{X^e} = \widehat{B}^e \# P^e$ for a matrix $\widehat{B}^e \in \mathbb{R}^{d_e \times \widehat{s}_e}$ and an $\widehat{s}_e$-dimensional probability measure $P^e$ with independent, mean-zero and unit-variance marginals $P_i^e$.
5: **end for**
6: Matching: Let $\widehat{\ell}$ be the maximal number such that there are signed permutation matrices $\overline{\{Q^e\}}_{e \in [m]}$ satisfying

$$d([(Q^e)^\top \# P^e]_i, [(Q^f)^\top \# P^f]_i) = 0$$

for all $i = 1, \dots, \widehat{\ell}$ and for all $f \neq e$. Let $\widehat{\mathcal{L}} = \{1, \dots, \widehat{\ell}\}$.
7: Construct the matrix $\widehat{B}$ and the tuple of probability measures $\widehat{P}$ given by

$$\widehat{B} = \begin{pmatrix} [\widehat{B}^1 Q^1]_{\widehat{\mathcal{L}}} & [\widehat{B}^1 Q^1]_{[\widehat{s}_1] \setminus \widehat{\mathcal{L}}} & & \\ \vdots & & \ddots & \\ [\widehat{B}^m Q^m]_{\widehat{\mathcal{L}}} & & & [\widehat{B}^m Q^m]_{[\widehat{s}_m] \setminus \widehat{\mathcal{L}}} \end{pmatrix} \text{ and } \widehat{P} = \begin{pmatrix} [(Q^1)^\top \# P^1]_{\widehat{\mathcal{L}}} \\ [(Q^1)^\top \# P^1]_{[\widehat{s}_1] \setminus \widehat{\mathcal{L}}} \\ \vdots \\ [(Q^m)^\top \# P^m]_{[\widehat{s}_m] \setminus \widehat{\mathcal{L}}} \end{pmatrix}.$$

8: **return** $(\widehat{\ell}, \widehat{B}, \widehat{P})$.

---

graph. That is, one may allow directed paths from shared to domain-specific latent nodes but not vice versa. For example, an additional edge $1 \to 4$ between the latent nodes $1$ and $4$ would be allowed in the graph in Figure 2. In this case, the dependency structure of the domains is still determined by the shared latent space. However, the structural assumption that there are no edges between shared and domain-specific latent nodes is made for identifiability of the shared latent graph in Section 4.

*Remark* 3.4. The computational complexity of Algorithm 1 depends on the complexity of the chosen linear ICA algorithm, to which we make $m$ calls. Otherwise, the dominant part is the matching in Line 6 with worst case complexity $\mathcal{O}(m \cdot \max_{e \in [m]} d_e^2)$, where we recall that $m$ is the number of domains and $d_e$ is the dimension of domain $e$.

In Appendix D we state a complete version of Algorithm 1 for the finite sample setting. In particular, we provide a method for the matching in Line 6 based on the two-sample Kolmogorov-Smirnov test. For finite samples, there might occur false discoveries, that is, distributions are matched that are actually not the same. With our method, we show that the probability of falsely discovering shared nodes shrinks exponentially with the number of domains.

## 4  Identifiability of the Causal Graph

We return to our goal of identifying the causal graph $\mathcal{G}_{\mathcal{L}} = (\mathcal{L}, D_{\mathcal{L}\mathcal{L}})$ among the shared latent nodes. By Theorem 3.1, we can identify the representation $(B, P)$ of $P_X \in \mathcal{M}(\mathcal{G}_m)$ from the marginal distributions. In particular, we recover the matrix $\widehat{B} = B\Psi$ for a signed permutation block matrix $\Psi \in \Pi$. Moreover, we know which columns correspond to the shared latent nodes. That is, we know that the submatrix $\widehat{B}_{\mathcal{L}}$ obtained by only considering the columns indexed by $\mathcal{L} = \widehat{\mathcal{L}} = [\ell]$ is equal to $B_{\mathcal{L}}\Psi_{\mathcal{L}}$, where $\Psi_{\mathcal{L}} \in SP(\ell)$.

**Problem 4.1.** Let $B \in \text{Im}(\phi_{\mathcal{G}_m})$ for an $m$-domain graph $\mathcal{G}_m$ with shared latent nodes $\mathcal{L}$. Given $\widehat{B}_{\mathcal{L}} = B_{\mathcal{L}}\Psi_{\mathcal{L}}$ with $\Psi_{\mathcal{L}}$ a signed permutation matrix, when is it possible to identify the graph $\mathcal{G}_{\mathcal{L}}$?

Recently, Xie et al. (2022) and Dai et al. (2022) show that, in the one-domain setting with independent additive noise, the latent graph can be identified if each latent variable has at least two pure children. We obtain a comparable result tailored to the multi-domain setup.

**Definition 4.2.** Let $\mathcal{G}_m = (\mathcal{H} \cup V, D)$ be an $m$-domain graph with shared latent nodes $\mathcal{L} \subseteq \mathcal{H}$. For $k \in \mathcal{L}$, we say that an observed node $v \in V$ is a *partial pure child* of $k$ if $\text{pa}(v) \cap \mathcal{L} = \{k\}$.

---

**Algorithm 2** IdentifySharedGraph

---

1: **Input:** Matrix $B^* \in \mathbb{R}^{|V| \times \ell}$.
2: **Output:** Parameter matrix $\widehat{A} \in \mathbb{R}^{\ell \times \ell}$.
3: Remove rows $B^*_{i,\mathcal{L}}$ from the matrix $B^*$ that are completely zero.
4: Find tuples $(i_k, j_k)_{k \in \mathcal{L}}$ with $i_k \neq j_k$ such that
      (i) $\text{rank}(B^*_{\{i_k, j_k\}, \mathcal{L}}) = 1$ for all $k \in \mathcal{L}$ and
      (ii) $\text{rank}(B^*_{\{i_k, i_q\}, \mathcal{L}}) = 2$ for all $k, q \in \mathcal{L}$ with $k \neq q$.
5: Let $I = \{i_1, \ldots, i_\ell\}$ and consider $B^*_{I, \mathcal{L}} \in \mathbb{R}^{\ell \times \ell}$.
6: Find two permutation matrices $R_1$ and $R_2$ such that $W = R_1 B^*_{I, \mathcal{L}} R_2$ is lower triangular.
7: Multiply each column of $W$ by the sign of its corresponding diagonal element. This yields a new matrix $\widetilde{W}$ with all diagonal elements positive.
8: Divide each row of $\widetilde{W}$ by its corresponding diagonal element. This yields a new matrix $\widetilde{W}'$ with all diagonal elements equal to one.
9: Compute $\widehat{A} = I - (\widetilde{W}')^{-1}$.
10: **return** $\widehat{A}$.

---

For a partial pure child $v \in V$, there may still be domain-specific latent nodes that are parents of $v$. Definition 4.2 only requires that there is exactly one parent that is in the set $\mathcal{L}$. This explains the name *partial* pure child; see Example B.2 in the Appendix for further elaboration.

W.l.o.g. we assume in this section that the shared latent nodes are *topologically ordered* such that $i \to j \in D_{\mathcal{L}\mathcal{L}}$ implies $i < j$ for all $i, j \in \mathcal{L}$. We further assume:

(C3) (Two partial pure children across domains.) For each shared latent node $k \in \mathcal{L}$, there exist two partial pure children.

(C4) (Rank faithfulness.) For any two subsets $Y \subseteq V$ and $W \subseteq \mathcal{L}$, we assume that

$$\text{rank}(B_{Y,W}) = \max_{B' \in \text{Im}(\phi_{\mathcal{G}_m})} \text{rank}(B'_{Y,W}).$$

The two partial pure children required in Condition (C3) may either be in distinct domains or in a single domain. This is a sparsity condition on the large mixing matrix $G$. In Appendix C we discuss that the identification of the joint latent graph is impossible without any sparsity assumptions. We conjecture that two partial pure children are not necessary, but we leave it open for future work to find a non-trivial necessary condition. Roughly speaking, we assume in Condition (C4) that no configuration of edge parameters coincidentally yields low rank. The set of matrices $B \in \text{Im}(\phi_{\mathcal{G}_m})$ that violates (C4) is a subset of measure zero of $\text{Im}(\phi_{\mathcal{G}_m})$ with respect to the Lebesgue measure. Note that our conditions do not impose constraints on the graph $\mathcal{G}_{\mathcal{L}}$. Our main tool to tackle Problem 4.1 will be the following lemma.

**Lemma 4.3.** *Let $B \in \text{Im}(\phi_{\mathcal{G}_m})$ for an $m$-domain graph $\mathcal{G}_m$. Suppose that Condition (C4) is satisfied and that there are no zero-rows in $B_{\mathcal{L}}$. Let $v, w \in V$. Then $\text{rank}(B_{\{v,w\},\mathcal{L}}) = 1$ if and only if there is a node $k \in \mathcal{L}$ such that both $v$ and $w$ are partial pure children of $k$.*

The condition on no zero-rows in Lemma 4.3 is needed since we always have $\text{rank}(B_{\{v,w\},\mathcal{L}}) \leq 1$ if one of the two rows is zero. However, this is no additional structural assumption since we allow zero-rows when identifying the latent graph; c.f. Algorithm 2. The lemma allows us to find partial pure children by testing ranks on the matrix $\widehat{B}_{\mathcal{L}}$. If $(i_1, j_1)$ and $(i_2, j_2)$ are partial pure children of two nodes in $\mathcal{L}$, we make sure that these two nodes are different by checking that $\text{rank}(B_{\{i_1, i_2\}, \mathcal{L}}) = 2$.

For a DAG $G = (V, D)$, we define $\mathcal{S}(G)$ to be the set of permutations on $|V|$ elements that are consistent with the DAG, i.e., $\sigma \in \mathcal{S}(G)$ if and only if $\sigma(i) < \sigma(j)$ for all edges $i \to j \in D$. The following result is the main result of this section.

**Theorem 4.4.** *Let $\widehat{B} = B\Psi$ with $B \in \text{Im}(\phi_{\mathcal{G}_m})$ and $\Psi \in \Pi$, and define $B^* = \widehat{B}_{\mathcal{L}}$ to be the input of Algorithm 2. Assume that Conditions (C3) and (C4) are satisfied, and let $\widehat{A}$ be the output of Algorithm 2. Then $\widehat{A} = Q_\sigma^\top A_{\mathcal{L}, \mathcal{L}} Q_\sigma$ for a signed permutation matrix $Q_\sigma$ with $\sigma \in \mathcal{S}(\mathcal{G}_{\mathcal{L}})$. Moreover, if $G_{vk} > 0$ for $G \in \mathbb{R}^{D_V \mathcal{H}}$ whenever $v$ is a pure child of $k$, then $Q_\sigma$ is a permutation matrix.*

Theorem 4.4 says that the graph $\mathcal{G}_\mathcal{L}$ can be recovered up to a permutation of the nodes that preserves the property that $i \to j$ implies $i < j$; see Remark 4.5. Since the columns of the matrix $\widehat{B}$ are not only permuted but also of different signs, we solve the sign indeterminacy column-wise in Line 7 before removing the scaling indeterminacy row-wise in Line 8. In case the coefficients of partial pure children are positive, this ensures that $Q_\sigma$ is a *permutation matrix* and we have no sign indeterminacy. In Appendix D we adapt Algorithm 2 for the empirical data setting, where we only have $\widehat{B}_\mathcal{L} \approx B_\mathcal{L}\psi_\mathcal{L}$.

*Remark* 4.5. Let $\widehat{A}$ be the output of Alg. 2. Then we construct the graph $\widehat{G}_\mathcal{L} = (\mathcal{L}, \widehat{D}_{\mathcal{L}\mathcal{L}})$ as the graph with edges $j \to i \in \widehat{D}_{\mathcal{L}\mathcal{L}}$ if and only if $\widehat{A}_{ij} \neq 0$. Condition (C4) ensures that $\widehat{G}_\mathcal{L}$ is equivalent to $\mathcal{G}_\mathcal{L}$ in the sense that there is a permutation $\sigma \in \mathcal{S}(\mathcal{G}_\mathcal{L})$ such that $\widehat{D}_{\mathcal{L}\mathcal{L}} = \{\sigma(i) \to \sigma(j) : i \to j \in D_{\mathcal{L}\mathcal{L}}\}$.

**Example 4.6.** As highlighted in the introduction, the unpaired multi-domain setup is motivated by applications from single-cell biology. For example, consider the domains of (i) gene expression and (ii) high-level phenotypic features extracted from imaging assays (e.g. McQuin et al., 2018). We argue that the requirement of two partial pure children is justifiable on such data as follows. The condition requires, for example, that for each shared latent variable, (i) the expression of some gene depends only upon that shared latent variable plus domain-specific latent variables, and (ii) one of the high-level phenotypic features depends only on the same latent feature plus domain-specific latent variables. Many genes have highly specialized functions, so (i) is realistic, and similarly many phenotypic features are primarily controlled by specific pathways, so (ii) is justified.

*Remark* 4.7. In Algorithm 2, we determine the rank of a matrix by Singular Value Decomposition, which has worst case complexity $\mathcal{O}(mn\min\{n,m\})$ for an $m \times n$ matrix. Since Line 4 is the dominant part, we conclude that the worst case complexity of Algorithm 2 is $\mathcal{O}(|V|^2 \cdot \ell)$.

# 5 Simulations

In this section we report on a small simulation study to illustrate the validity of our adapted algorithms for finite samples (detailed in Appendix D). We emphasize that this should only serve as a proof of concept as the focus of our work lies on identifiability. In future work one may develop more sophisticated methods; c.f. Appendix G. The adapted algorithms have a hyperparameter $\gamma$, which is a threshold on singular values to determine the rank of a matrix. In our simulations we use $\gamma = 0.2$.

**Data Generation.** In each experiment we generate 1000 random models with $\ell = 3$ shared latent nodes. We consider different numbers of domains $m \in \{2, 3\}$ and assume that there are $|I_e| = 2$ domain-specific latent nodes for each domain. The dimensions are given by $d_e = d/m$ for all $e \in [m]$ and $d = 30$. We sample the $m$-domain graph $\mathcal{G}_m$ on the shared latent nodes as follows. First, we sample the graph $\mathcal{G}_\mathcal{L}$ from an Erdős-Rényi model with edge probability 0.75 and assume that there are no edges between other latent nodes, that is, between $\mathcal{L}$ and $\mathcal{H} \setminus \mathcal{L}$ and within $\mathcal{H} \setminus \mathcal{L}$. Then we fix two partial pure children for each shared latent node $k \in \mathcal{L}$ and collect them in the set $W$. The remaining edges from $\mathcal{L}$ to $V \setminus W$ and from $\mathcal{H}$ to $V$ are sampled from an Erdős-Rényi model with edge probability 0.9. Finally, the (nonzero) entries of $G$ and $A$ are sampled from $\text{Unif}(\pm[0.25, 1])$. The distributions of the error variables are specified in Appendix E. For simplicity, we assume that the sample sizes coincide, that is, $n_e = n$ for all $e \in [m]$, and consider $n \in \{1000, 2500, 5000, 10000, 25000\}$.

**Results.** First, we plot the average number of shared nodes $\widehat{\ell}$ in our experiments in Figure 3 (a). Especially for low sample sizes, we see that fewer shared nodes are detected with more domains. However, by inspecting the error bars we also see that the probability of detecting too many nodes $\widehat{\ell} > \ell$ decreases drastically when considering 3 instead of 2 domains. This suggests that the number of falsely detected shared nodes is very low, as expected by Theorem D.3. Our findings show that more domains lead to a more conservative discovery of shared nodes, but whenever a shared node is determined this is more certain. Moreover, we measure the error in estimating $\widehat{B}_{\widehat{\mathcal{L}}}$ in Figure 3 (b), that is, the error in the "shared" columns. We take

$$\text{score}_B(\widehat{B}_{\widehat{\mathcal{L}}}) = \begin{cases} \min_{\Psi \in SP(\ell)} \beta_{\ell,\widehat{\ell}}^{-1/2}\|\widehat{B}_{\widehat{\mathcal{L}}} - [B_\mathcal{L}\Psi]_{\widehat{\mathcal{L}}}\|_\mathcal{F} & \text{if } \widehat{\ell} \leq \ell, \\ \min_{\Psi \in SP(\widehat{\ell})} \beta_{\ell,\widehat{\ell}}^{-1/2}\|[\widehat{B}_{\widehat{\mathcal{L}}}\Psi]_\mathcal{L} - B_\mathcal{L}\|_\mathcal{F} & \text{if } \widehat{\ell} > \ell, \end{cases}$$

where $\|\cdot\|_\mathcal{F}$ denotes the Frobenius norm and $\beta_{\ell,\widehat{\ell}} = \min\{\ell, \widehat{\ell}\} \cdot \sum_{e=1}^m d_e$ denotes the number of entries of the matrix over which the norm is taken. In the cases $\ell = \widehat{\ell}$, we also measure the

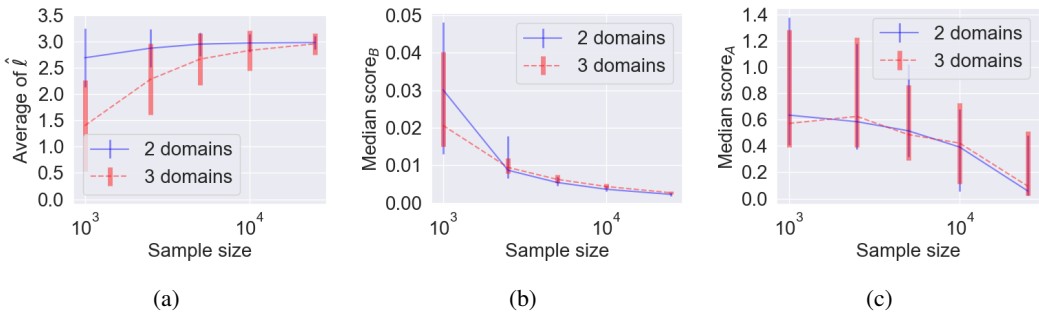

Figure 3: **Results.** Logarithmic scale on the $x$-axis. Error bars in (a) are one standard deviation of the mean and in (b) and (c) they are the interquartile range.

performance of recovering the shared latent graph $\mathcal{G}_\mathcal{L}$ in Figure 3 (c) by taking

$$\text{score}_A(\widehat{A}) = \min_{Q_\sigma \in SP(\ell) \text{ s.t. } \sigma \in S(\mathcal{G}_\mathcal{L})} \frac{1}{\ell} \|Q_\sigma^\top \widehat{A} Q_\sigma - A_{\mathcal{L},\mathcal{L}}\|_\mathcal{F}.$$

As expected, the median estimation errors for $B_\mathcal{L}$ and $A_{\mathcal{L},\mathcal{L}}$ decrease with increasing sample size. In Appendix F we provide additional simulations with larger $\ell$. Moreover, we consider setups where we violate specific assumptions, such as pairwise different distributions (C1) and two partial pure children (C3). The results emphasize that the conditions are necessary for the algorithms provided. The computations were performed on a single thread of an Intel Xeon Gold 6242R processor (3.1 GHz), with a total computation time of 12 hours for all simulations presented in this paper (including Appendix).

## 6 Discussion

This work introduces the problem of causal representation learning from *unpaired* multi-domain observations, in which multiple domains provide complementary information about a set of shared latent nodes that are the causal quantities of primary interest. For this problem, we laid out a setting in which we can provably identify the causal relations among the shared latent nodes. To identify the desired causal structure, we proposed a two-step approach where we first make use of linear ICA in each domain separately and match the recovered error distributions to identify shared nodes and the joint distribution of the domains. In the second step, we identify the causal structure among the shared latent variables by testing rank deficiencies in the "overall mixing matrix" $B$. To the best of our knowledge, our guarantees are the first principled identifiability results for shared causal representations in a general, unpaired multi-domain setting.

We proposed algorithms for recovering the joint distribution and the shared latent space making our proofs constructive. While our focus is on identifiability guarantees, we show in Appendix D how our proofs give rise to algorithms for the finite sample setting. Moreover, we propose a method to match approximate error distributions and show that the probability of falsely discovering shared nodes decreases exponentially in the number of domains. Our work opens up numerous directions for future work as we discuss in Appendix G.

## Acknowledgments and Disclosure of Funding

This project was initiated while the first author was a visitor at the Eric and Wendy Schmidt Center of the Broad Institute of MIT and Harvard. The project has received funding from the European Research Council (ERC) under the European Union's Horizon 2020 research and innovation programme (grant agreement No 883818), NCCIH/NIH (1DP2AT012345), ONR (N00014-22-1-2116), DOE-ASCR (DE-SC0023187), NSF (DMS-1651995), the MIT-IBM Watson AI Lab, and a Simons Investigator Award. Nils Sturma acknowledges support by the Munich Data Science Institute at the Technical University of Munich via the Linde/MDSI PhD Fellowship program. Chandler Squires was partially supported by an NSF Graduate Research Fellowship.

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

# A Proofs

*Proof of Theorem 3.1.* Let $P_X \in \mathcal{M}(\mathcal{G}_m)$ for an $m$-domain graph $\mathcal{G}_m = (\mathcal{H} \cup V, D)$ with shared latent nodes $\mathcal{L} = [\ell]$ and representation $(B, P)$. By Condition (C1) the measure $P$ has independent, non-degenerate, non-symmetric marginals $P_i$, $i \in \mathcal{H}$ with mean zero and variance one. Moreover, since $B \in \mathrm{Im}(\phi_{\mathcal{G}_m})$, we have $B = G(I - A)^{-1}$ for matrices $G \in \mathbb{R}^{D_V \mathcal{H}}$ and $A \in \mathbb{R}^{D_\mathcal{H} \mathcal{H}}$.

Fix one domain $e \in [m]$. Recall that we denote by $S_e = \mathrm{pa}(V_e) = \mathcal{L} \cup I_e$ the set of latent parents in domain $e$. Define the matrix

$$B^e := G_{V_e, S_e}[(I - A)^{-1}]_{S_e, S_e} = G^e[(I - A)^{-1}]_{S_e, S_e},$$

and observe that we can write $P_{X^e} = B_{V_e, \mathcal{H}} \# P = B^e \# P_{S_e}$. This is due to the fact that $G_{V_e, \mathcal{H} \setminus S_e} = 0$ and $[(I - A)^{-1}]_{S_e, \mathcal{H} \setminus S_e} = 0$ by the definition of an $m$-domain graph.

In particular, the equality $P_{X^e} = B^e \# P_{S_e}$ shows that the representation in Line 4 of Algorithm 1 exists. Now, we show that it is unique up to signed permutation by applying results on identifiability of linear ICA. Since $G^e$ has full column rank by Condition (C2) and $[(I - A)^{-1}]_{S_e, S_e}$ is invertible, the matrix $B^e$ also has full column rank. Let $P_{X^e} = \widehat{B}^e \# P^e$ be any representation, where $\widehat{B}^e \in \mathbb{R}^{d_e \times \widehat{s}_e}$ and $P^e$ is an $\widehat{s}_e$-dimensional probability measure with independent, non-degenerate marginals $P_i^e$. Due to Condition (C1), all probability measures $P_i$ are non-Gaussian and non-degenerate and therefore we have by Eriksson and Koivunen (2004, Theorem 3 and 4) the identities

$$\widehat{B}^e = B^e R^e \Lambda^e \quad \text{and} \quad P^e = \Lambda^e (R^e)^\top \# P_{S_e}, \tag{3}$$

where $\Lambda^e$ is an $s_e \times s_e$ diagonal matrix with nonzero entries and $R^e$ is an $s_e \times s_e$ permutation matrix. In particular, we have $\widehat{s}_e = s_e$, which means that $\widehat{B}^e \in \mathbb{R}^{d_e \times s_e}$ and that $P^e$ is an $s_e$-dimensional probability measure. Line 4 also requires that each marginal $P_i^e$ has unit variance. This removes the scaling indeterminacy in (3) and we have

$$\widehat{B}^e = B^e R^e D^e \quad \text{and} \quad P^e = D^e (R^e)^\top \# P_{S_e},$$

where $D^e$ is a diagonal matrix with entries $D_{ii}^e \in \{\pm 1\}$. In particular, this means that the distributions $P^e$ and $P_{S_e}$ coincide up to permutation and sign of the marginals.

The matching in Line 6 identifies which components of $P^e$ are shared. By Condition (C1), two components of different domains $P_i^e$ and $P_j^f$ are shared if and only if they coincide up to sign, that is, if and only if $d(P_i^e, P_j^f) = 0$ or $d(P_i^e, -P_j^f) = 0$. If their distribution coincide up to sign, than either $d(P_i^e, P_j^f) = 0$ or $d(P_i^e, -P_j^f) = 0$ but not both since Condition (C1) requires the distribution of the error variables to be non-symmetric. We conclude that in each domain $e \in [m]$ there exists an $s_e \times s_e$ signed permutation matrix $Q^e$ such that

$$d([(Q^e)^\top \# P^e]_i, [(Q^f)^\top \# P^f]_i) = 0 \tag{4}$$

for all $i = 1, \ldots, \widehat{\ell}$ and for all $f \neq e$. In particular, $\widehat{\ell} = \ell$ and $\widehat{\mathcal{L}} = \mathcal{L}$.

It remains to show that $\widehat{B} = B\Psi$ and $\widehat{P} = \Psi^\top \# P$ for a signed permutation block matrix $\Psi \in \Pi$. By Equation (4), the distributions $[(Q^e)^\top \# P^e]_\mathcal{L}$ and $[(Q^e)^\top \# P^e]_\mathcal{L}$ coincide, which means that

$$(Q^e)^\top \# P^e = (Q^e)^\top D^e (R^e)^\top \# P_{S_e} = \begin{pmatrix} \Psi_\mathcal{L}^\top & 0 \\ 0 & \Psi_{I_e}^\top \end{pmatrix} \# \begin{pmatrix} P_\mathcal{L} \\ P_{I_e} \end{pmatrix}, \tag{5}$$

where $\Psi_\mathcal{L}$ is an $\ell \times \ell$ signed permutation matrix and $\Psi_{I_e}$ is an $|I_e| \times |I_e|$ signed permutation matrix. Importantly, the matrix $\Psi_\mathcal{L}^\top$ does not depend on the domain $e \in [m]$. Hence, the matrix $\Phi^e := R^e D^e Q^e$ is a signed permutation matrix with block structure as in Equation (5). Moreover, we have

$$\widehat{B}^e Q^e = B^e R^e D^e Q^e = B^e \Phi^e = \left( B_\mathcal{L}^e \Psi_\mathcal{L} \mid B_{[s_e] \setminus \mathcal{L}}^e \Psi_{I_e} \right),$$

which means that the matrix $\widehat{B}$ can be factorized as

$$\widehat{B} = \begin{pmatrix} [\widehat{B}^1 Q^1]_{\widehat{\mathcal{L}}} & [\widehat{B}^1 Q^1]_{[\widehat{s}_1]\setminus\widehat{\mathcal{L}}} & & \\ \vdots & & \ddots & \\ [\widehat{B}^m Q^m]_{\widehat{\mathcal{L}}} & & & [\widehat{B}^m Q^m]_{[\widehat{s}_m]\setminus\widehat{\mathcal{L}}} \end{pmatrix}$$

$$= \begin{pmatrix} B^1_{\mathcal{L}}\Psi_{\mathcal{L}} & B^1_{[s_1]\setminus\mathcal{L}}\Psi_{I_1} & & \\ \vdots & & \ddots & \\ B^m_{\mathcal{L}}\Psi_{\mathcal{L}} & & & B^m_{[s_m]\setminus\mathcal{L}}\Psi_{I_m} \end{pmatrix}$$

$$= \begin{pmatrix} B^1_{\mathcal{L}} & B^1_{[s_1]\setminus\mathcal{L}} & & \\ \vdots & & \ddots & \\ B^m_{\mathcal{L}} & & & B^m_{[s_m]\setminus\mathcal{L}} \end{pmatrix} \cdot \begin{pmatrix} \Psi_{\mathcal{L}} & & & \\ & \Psi_{I_1} & & \\ & & \ddots & \\ & & & \Psi_{I_m} \end{pmatrix} = B \cdot \Psi,$$

where $\Psi \in \Pi$. Similarly, we have for all $e \in [m]$,

$$(Q^e)^\top \# P^e = (\Phi^e)^\top \# P_{S_e} = \begin{pmatrix} (\Psi_{\mathcal{L}})^\top \# P_{\mathcal{L}} \\ (\Psi_{I_e})^\top \# P_{I_e} \end{pmatrix}.$$

We conclude that

$$\widehat{P} = \begin{pmatrix} [(Q^1)^\top \# P^1]_{\widehat{\mathcal{L}}} \\ [(Q^1)^\top \# P^1]_{[\widehat{s}_1]\setminus\widehat{\mathcal{L}}} \\ \vdots \\ [(Q^m)^\top \# P^m]_{[\widehat{s}_m]\setminus\widehat{\mathcal{L}}} \end{pmatrix} = \begin{pmatrix} (\Psi_{\mathcal{L}})^\top \# P_{\mathcal{L}} \\ (\Psi_{I_1})^\top \# P_{I_1} \\ \vdots \\ (\Psi_{I_m})^\top \# P_{I_m} \end{pmatrix}$$

$$= \begin{pmatrix} \Psi_{\mathcal{L}} & & & \\ & \Psi_{I_1} & & \\ & & \ddots & \\ & & & \Psi_{I_m} \end{pmatrix}^\top \# \begin{pmatrix} P_{\mathcal{L}} \\ P_{I_1} \\ \vdots \\ P_{I_m} \end{pmatrix} = \Psi^\top \# P.$$

$\square$

Before proving Lemma 4.3 and Theorem 4.4 we fix some notation. Let $\mathcal{G}_m = (V \cup \mathcal{H}, D)$ be an $m$-domain graph. We denote by $\operatorname{anc}(v) = \{k \in \mathcal{H} : \text{there is a directed path } k \to \cdots \to v \text{ in } \mathcal{G}_m\}$ the ancestors of a node $v \in V$. For subsets $W \subseteq V$, we denote $\operatorname{anc}(W) = \bigcup_{v \in W} \operatorname{anc}(w)$. Moreover, for $\mathcal{L} \subseteq \mathcal{H}$ and $v \in V$, we write shortly $\operatorname{pa}_{\mathcal{L}}(v) = \operatorname{pa}(v) \cap \mathcal{L}$.

*Proof of Lemma 4.3.* Let $B \in \operatorname{Im}(\phi_{\mathcal{G}_m})$. Then we can write $B = G \cdot (I - A)^{-1}$ with

$$G = \begin{pmatrix} G_{V_1,\mathcal{L}} & G_{V_1,I_1} & & \\ \vdots & & \ddots & \\ G_{V_m,\mathcal{L}} & & & G_{V_m,I_m} \end{pmatrix}.$$

Moreover, observe that, by the definition of an $m$-domain-graph, the matrix $B_{V,\mathcal{L}}$ factorizes as

$$B_{V,\mathcal{L}} = G_{V,\mathcal{L}}[(I - A)^{-1}]_{\mathcal{L},\mathcal{L}}.$$

Now, suppose that $i$ and $j$ are partial pure children of a fixed node $k \in \mathcal{L}$. Then $\operatorname{pa}_{\mathcal{L}}(i) = \{k\} = \operatorname{pa}_{\mathcal{L}}(j)$. In particular, the only entry that may be nonzero in the row $G_{i,\mathcal{L}}$ is given by $G_{ik}$ and the only entry that may be nonzero in the row $G_{j,\mathcal{L}}$ is given by $G_{jk}$. Thus, we have

$$B_{i,\mathcal{L}} = \sum_{q \in \mathcal{L}} G_{iq}[(I - A)^{-1}]_{q,\mathcal{L}} = G_{ik}[(I - A)^{-1}]_{k,\mathcal{L}}.$$

Similarly, it follows that $B_{j,\mathcal{L}} = G_{jk}[(I - A)^{-1}]_{k,\mathcal{L}}$. This means that the row $B_{j,\mathcal{L}}$ is a multiple of the row $B_{i,\mathcal{L}}$ and we conclude that $\operatorname{rank}(B_{\{i,j\},\mathcal{L}}) \leq 1$. Equality holds due to the faithfulness condition (C4) which implies that $B_{ik} \neq 0$ and $B_{jk} \neq 0$, i.e., $B_{\{i,j\},\mathcal{L}}$ is not the null matrix.

For the other direction suppose that $\text{rank}(B_{\{i,j\},\mathcal{L}}) = 1$. By applying the Lindström-Gessel-Viennot Lemma (Gessel and Viennot, 1985; Lindström, 1973) equivalently as in Dai et al. (2022, Theorem 1 and 2), it can be seen that

$$\text{rank}(B_{\{i,j\},\mathcal{L}}) \leq \min\{|S| : S \text{ is a vertex cut from anc}(\mathcal{L}) \text{ to } \{i,j\}\}, \qquad (6)$$

where $S$ is a vertex cut from $\text{anc}(\mathcal{L})$ to $\{i,j\}$ if and only if there exists no directed path in $\mathcal{G}_m$ from $\text{anc}(\mathcal{L})$ to $\{i,j\}$ without passing through $S$. Moreover, equality holds in (6) for generic (almost all) choices of parameters. Since we assumed rank faithfulness in Condition (C4) we exclude cases where the inequality is strict and therefore have equality. By the definition of an $m$-domain graph we have that $\text{anc}(\mathcal{L}) = \mathcal{L}$. Thus, if $\text{rank}(B_{\{i,j\},\mathcal{L}}) = 1$, there must be a single node $k \in \mathcal{L}$ such that $\{k\}$ is a vertex cut from $\mathcal{L}$ to $\{i,j\}$. But it follows that $i$ and $j$ have to be partial pure children of $k$ by the definition of an $m$-domain graph and by using the assumption that there are no zero-rows in $B_\mathcal{L}$. $\quad\square$

To prove Theorem 4.4 we need the following auxiliary lemma.

**Lemma A.1.** *Let $G = (V, D)$ be a DAG with topologically ordered nodes $V = [p]$ and let $M$ be a lower triangular matrix with entries $M_{ii} \neq 0$ for all $i = 1, \ldots, p$ and $M_{ij} \neq 0$ if and only if there is a directed path $j \to \cdots \to i$ in $G$. Let $Q_{\sigma_1}$ and $Q_{\sigma_2}$ be permutation matrices. Then the matrix $Q_{\sigma_1} M Q_{\sigma_2}$ is lower triangular if and only if $\sigma_2 = \sigma_1^{-1}$ and $\sigma_2 \in \mathcal{S}(G)$.*

*Proof of Lemma A.1.* By the definition of a permutation matrix, we have

$$[Q_{\sigma_1} M Q_{\sigma_2}]_{ij} = M_{\sigma_1(i)\sigma_2^{-1}(j)} \quad \text{or, equivalently,} \quad [Q_{\sigma_1} M Q_{\sigma_2}]_{\sigma_1^{-1}(i)\sigma_2(j)} = M_{ij}. \qquad (7)$$

First, suppose that $\sigma_2 = \sigma_1^{-1}$ and $\sigma_2 \in \mathcal{S}(G)$, and let $i, j \in [p]$ such that $\sigma_2(i) < \sigma_2(j)$. Then, by the definition of $\mathcal{S}(G)$, there is no directed path $j \to \cdots \to i$ in the graph $G$ and therefore we have $M_{ij} = 0$. But this means that $[Q_{\sigma_1} M Q_{\sigma_2}]_{\sigma_2(i)\sigma_2(j)} = 0$ and we conclude that the matrix $Q_{\sigma_1} M Q_{\sigma_2}$ is lower triangular.

Now, assume that $Q_{\sigma_1} M Q_{\sigma_2}$ is lower triangular, where $\sigma_1$ and $\sigma_2$ are arbitrary permutations on the set $[p]$. Since $M$ has no zeros on the diagonal, we have $M_{ii} = [Q_{\sigma_1} M Q_{\sigma_2}]_{\sigma_1^{-1}(i)\sigma_2(i)} \neq 0$ for all $i = 1, \ldots, p$. It follows that $\sigma_1^{-1}(i) \geq \sigma_2(i)$ for all $i = 1, \ldots, p$ because $Q_{\sigma_1} M Q_{\sigma_2}$ is lower triangular. But this is only possible if the permutations coincide on all elements, i.e., we have $\sigma_2 = \sigma_1^{-1}$. It remains to show that $\sigma_2 = \sigma_1^{-1} \in \mathcal{S}(G)$. For any edge $j \to i \in D$ we have that $M_{ij} \neq 0$. Recalling Equation (7) this means that $[Q_{\sigma_1} M Q_{\sigma_2}]_{\sigma_2(i)\sigma_2(j)} \neq 0$. But since $Q_{\sigma_1} M Q_{\sigma_2}$ is lower triangular this can only be the case if $\sigma_2(j) < \sigma_2(i)$ which proves that $\sigma_2 \in \mathcal{S}(G)$. $\quad\square$

*Proof of Theorem 4.4.* Each latent node in $\mathcal{L}$ has two partial pure children by Condition (C3). After removing zero-rows in Line 3 of Algorithm 2 it holds by Lemma 4.3 that $\text{rank}(B^*_{\{i,j\},\mathcal{L}}) = 1$ if and only if there is a latent node in $\mathcal{L}$ such that $i$ and $j$ are both partial pure children of that latent node. Hence, each tuple $(i_k, j_k)_{k \in \mathcal{L}}$ in Line 4 of Algorithm 2 consists of two partial pure children of a certain latent node. The requirement $\text{rank}(B^*_{\{i_k,i_q\},\mathcal{L}}) = 2$ ensures that each pair of partial pure children has a different parent.

By the definition of an $m$-domain-graph and the fact that $B^* = \widehat{B}_\mathcal{L}$, for $I = \{i_1, \ldots, i_\ell\}$, we have the factorization

$$B^*_{I,\mathcal{L}} = B_{I,\mathcal{L}} \Psi_\mathcal{L} = G_{I,\mathcal{L}} (I - A)^{-1}_{\mathcal{L},\mathcal{L}} \Psi_\mathcal{L} = G_{I,\mathcal{L}} (I - A_{\mathcal{L},\mathcal{L}})^{-1} \Psi_\mathcal{L}, \qquad (8)$$

where $G \in \mathbb{R}^{D_V \times \mathcal{H}}$, $A \in \mathbb{R}^{D_\mathcal{H} \times \mathcal{H}}$ and $\Psi_\mathcal{L}$ is a signed permutation matrix. Let $Q_1$ and $Q_2$ be permutation matrices and let $\Lambda$ be a diagonal matrix with non-zero diagonal elements and let $D$ be a diagonal matrix with entries in $\{\pm 1\}$. Then we can rewrite Equation (8) as

$$B^*_{I,\mathcal{L}} = Q_1 \underbrace{\Lambda (I - A_{\mathcal{L},\mathcal{L}})^{-1} D}_{=:M} Q_2.$$

Now, we apply Lemma A.1. Since we assume throughout Section 4 that the nodes $\mathcal{L}$ are topologically ordered, the matrix $M$ is lower triangular with no zeros on the diagonal. Moreover, by Condition (C4) we have $M_{ij} \neq 0$ if and only if there is a directed path $j \to \cdots \to i$ in $\mathcal{G}_\mathcal{L}$. In Line 6 we find other permutation matrices $R_1$ and $R_2$ such that

$$W = R_1 B^*_{I,\mathcal{L}} R_2 = (R_1 Q_1) M (Q_2 R_2)$$

is lower triangular. Now, define the permutation matrices $Q_{\sigma_1} = R_1 Q_1$ and $Q_{\sigma_2} = Q_2 R_2$. Then we have by Lemma A.1 that $Q_{\sigma_1} = Q_{\sigma_2}^\top$ and that $\sigma_2 \in \mathcal{S}(\mathcal{G}_\mathcal{L})$. Hence, the matrix $W$ factorizes as

$$W = Q_{\sigma_2}^\top M Q_{\sigma_2} = Q_{\sigma_2}^\top \Lambda (I - A_{\mathcal{L},\mathcal{L}})^{-1} D Q_{\sigma_2} = \widetilde{\Lambda} Q_{\sigma_2}^\top (I - A_{\mathcal{L},\mathcal{L}})^{-1} Q_{\sigma_2} \widetilde{D},$$

where $\widetilde{\Lambda}$ and $\widetilde{D}$ are diagonal matrices with the entries given by permutations of the entries of $\Lambda$ and $D$. Lines 7 and 8 address the scaling and sign matrices $\widetilde{\Lambda}$ and $\widetilde{D}$. In particular, we have that $\widetilde{W}' = D' Q_{\sigma_2}^\top (I - A_{\mathcal{L},\mathcal{L}})^{-1} Q_{\sigma_2} D'$ for another diagonal matrix $D'$ with entries in $\{\pm 1\}$, since each entry on the diagonal of $\widetilde{W}'$ is equal to 1. Thus, we have

$$\begin{aligned}
\widehat{A} &= I - (\widetilde{W}')^{-1} \\
&= I - (D' Q_{\sigma_2}^\top (I - A_{\mathcal{L},\mathcal{L}})^{-1} Q_{\sigma_2} D')^{-1} \\
&= I - D' Q_{\sigma_2}^\top (I - A_{\mathcal{L},\mathcal{L}}) Q_{\sigma_2} D' \\
&= D' Q_{\sigma_2}^\top A_{\mathcal{L},\mathcal{L}} Q_{\sigma_2} D'.
\end{aligned}$$

Since $Q_{\sigma_2} D'$ is a signed permutation matrix with $\sigma_2 \in \mathcal{S}(\mathcal{G}_\mathcal{L})$, the first part of the theorem is proved. If $G_{vk} > 0$ whenever $v$ is a pure child of $k$, the matrix $\widetilde{\Lambda}$ only has positive entries which means that $D'$ is equal to the identity matrix. This proves the second part. $\qquad\square$

## B  Additional Examples

The graph in Figure 4 is an $m$-domain graph corresponding to the compact version in Figure 2 in the main paper.

**Example B.1.** Consider the $m$-domain graph in Figure 4. The linear structural equation model among the latent variables is determined by lower triangular matrices of the form

$$A = \begin{pmatrix} 0 & 0 & 0 & 0 & 0 \\ a_{21} & 0 & 0 & 0 & 0 \\ 0 & 0 & 0 & 0 & 0 \\ 0 & 0 & a_{43} & 0 & 0 \\ 0 & 0 & 0 & 0 & 0 \end{pmatrix}.$$

Moreover, the domain-specific mixing matrices $G^e$ are of the form

$$G^1 = \begin{pmatrix} g_{11}^1 & g_{12}^1 & g_{13}^1 & 0 \\ g_{21}^1 & 0 & g_{23}^1 & 0 \\ 0 & g_{32}^1 & g_{33}^1 & g_{34}^1 \\ g_{41}^1 & g_{42}^1 & g_{43}^1 & g_{44}^1 \end{pmatrix} \quad \text{and} \quad G^2 = \begin{pmatrix} g_{11}^2 & 0 & g_{13}^2 \\ g_{21}^2 & g_{22}^2 & g_{23}^2 \\ 0 & g_{32}^2 & 0 \\ g_{41}^2 & g_{42}^2 & g_{43}^2 \\ g_{51}^2 & g_{52}^2 & 0 \end{pmatrix}.$$

Since the shared latent nodes are given by $\mathcal{L} = \{1, 2\}$, we have

$$G = \begin{pmatrix} g_{11}^1 & g_{12}^1 & g_{13}^1 & 0 & 0 \\ g_{21}^1 & 0 & g_{23}^1 & 0 & 0 \\ 0 & g_{32}^1 & g_{33}^1 & g_{34}^1 & 0 \\ g_{41}^1 & g_{42}^1 & g_{43}^1 & g_{44}^1 & 0 \\ g_{11}^2 & 0 & 0 & 0 & g_{13}^2 \\ g_{21}^2 & g_{22}^2 & 0 & 0 & g_{23}^2 \\ 0 & g_{32}^2 & 0 & 0 & 0 \\ g_{41}^2 & g_{42}^2 & 0 & 0 & g_{43}^2 \\ g_{51}^2 & g_{52}^2 & 0 & 0 & 0 \end{pmatrix}$$

and

$$B = G \cdot (I - A)^{-1} = \begin{pmatrix} a_{21} g_{12}^1 + g_{11}^1 & g_{12}^1 & g_{13}^1 & 0 & 0 \\ g_{21}^1 & 0 & g_{23}^1 & 0 & 0 \\ a_{21} g_{32}^1 & g_{32}^1 & a_{43} g_{34}^1 + g_{33}^1 & g_{34}^1 & 0 \\ a_{21} g_{42}^1 + g_{41}^1 & g_{42}^1 & a_{43} g_{44}^1 + g_{43}^1 & g_{44}^1 & 0 \\ g_{11}^2 & 0 & 0 & 0 & g_{13}^2 \\ a_{21} g_{22}^2 + g_{21}^2 & g_{22}^2 & 0 & 0 & g_{23}^2 \\ a_{21} g_{32}^2 & g_{32}^2 & 0 & 0 & 0 \\ a_{21} g_{42}^2 + g_{41}^2 & g_{42}^2 & 0 & 0 & g_{43}^2 \\ a_{21} g_{52}^2 + g_{51}^2 & g_{52}^2 & 0 & 0 & 0 \end{pmatrix}.$$

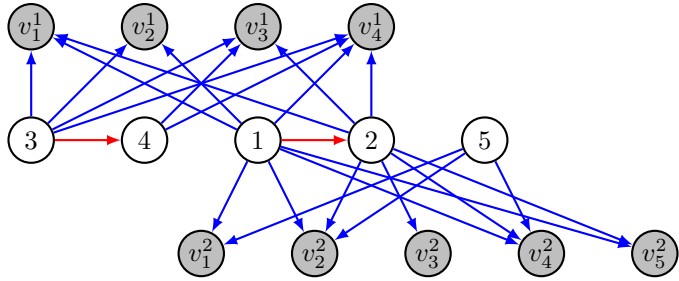

Figure 4: A 2-domain graph with 5 latent nodes and dimensions of the observed domains given by $|V_1| = d_1 = 4$ and $|V_2| = d_2 = 5$. We denote $V_e = \{v_1^e, \ldots, v_{d_e}^e\}$, that is, the superscript indicates the domain a node belongs to.

**Example B.2.** Consider the $m$-domain graph in Figure 4. The partial pure children of node $1 \in \mathcal{L}$ are given by $\{v_2^1, v_1^2\}$ and the partial pure children of $2 \in \mathcal{L}$ are given by $\{v_3^1, v_3^2\}$. Moreover, by continuing Example B.1, we have that the matrix $B_{\mathcal{L}}$ is given by

$$B_{\mathcal{L}} = \begin{pmatrix} a_{21}g_{12}^1 + g_{11}^1 & g_{12}^1 \\ g_{21}^1 & 0 \\ a_{21}g_{32}^1 & g_{32}^1 \\ a_{21}g_{42}^1 + g_{41}^1 & g_{42}^1 \\ g_{11}^2 & 0 \\ a_{21}g_{22}^2 + g_{21}^2 & g_{22}^2 \\ a_{21}g_{32}^2 & g_{32}^2 \\ a_{21}g_{42}^2 + g_{41}^2 & g_{42}^2 \\ a_{21}g_{52}^2 + g_{51}^2 & g_{52}^2 \end{pmatrix}$$

It is easy to see that the two submatrices

$$\begin{pmatrix} g_{21}^1 & 0 \\ g_{11}^2 & 0 \end{pmatrix} \quad \text{and} \quad \begin{pmatrix} a_{21}g_{32}^1 & g_{32}^1 \\ a_{21}g_{32}^2 & g_{32}^2 \end{pmatrix}$$

have rank one. The first matrix corresponds to the partial pure children $\{v_2^1, v_1^2\}$ in the graph in Figure 4 while the second matrix correspond to the partial pure children $\{v_3^1, v_3^2\}$. Note that the rank of any other $2 \times 2$ submatrix is generically (i.e., almost surely) equal to 2.

## C   Discussion of the Assumptions

In this section, we discuss aspects of Conditions (C1)-(C3) that allow for identifiability. In particular, we discuss the necessity of pairwise different and non-Gaussian error distributions if one is not willing to make further assumptions. Moreover, we elaborate on the sparsity conditions on the mixing matrix and explain why *some* sparsity assumption is necessary.

**Pairwise Different Error Distributions.** Given any two potentially different $m$-domain graphs $\mathcal{G}_m = (\mathcal{H} \cup V, D)$ and $\widetilde{\mathcal{G}}_m = (\widetilde{\mathcal{H}} \cup V, \widetilde{D})$, identifiability of the joint distribution in multi-domain causal representation models means that

$$B_{V_e,S_e} \# P_{S_e} = \widetilde{B}_{V_e,\widetilde{S}_e} \# \widetilde{P}_{\widetilde{S}_e} \text{ for all } e \in [m] \implies B \# P = \widetilde{B} \# \widetilde{P} \tag{9}$$

for any representation $(B, P)$ of a distribution in $\mathcal{M}(\mathcal{G}_m)$ and any representation $(\widetilde{B}, \widetilde{P})$ of a distribution in $\mathcal{M}(\widetilde{\mathcal{G}}_m)$, where the matrices $B_{V_e,S_e}$ and $\widetilde{B}_{V_e,\widetilde{S}_e}$ have full column rank for all $e \in [m]$. The left-hand side says that the marginal distributions in each domain are equal, while the right-hand side says that the joint distributions are equal. If there are $m$-domain graphs, such that the left-hand sides holds but the right-hand side is violated, then we say that the joint distribution is *not identifiable*.

We assume in this section that the marginal error distributions $P_i, i \in \mathcal{H}$ are non-Gaussian and have unit variance, but are not necessarily pairwise different or non-symmetric. Then the right-hand side holds if and only if the number of shared latent nodes in each graph is equal, i.e., $\ell = \widetilde{\ell}$, and there is a signed permutation matrix $\Psi$ such that $B = \widetilde{B}\Psi$ and $P = \Psi^\top \# \widetilde{P}$. Here, the matrix $\Psi$ does

not necessarily have a block structure. The equivalence is implied by the identifiability of the usual, one-domain linear ICA ( see, e.g., Buchholz et al. (2022)) together with the fact that for $\ell \neq \widetilde{\ell}$, we have $|\mathcal{H}| \neq |\widetilde{\mathcal{H}}|$ and, therefore, the distributions on the right-hand have support over different dimensional subspaces.

Theorem 3.1 shows that assumptions (C1) and (C2) are sufficient for identifiability of the joint distribution. In particular, we show that they imply identifiability in a stronger sense, namely, that it follows from the left-hand side that $\ell = \widetilde{\ell}$ and $B = \widetilde{B}\Psi$ and $P = \Psi^\top \# \widetilde{P}$ for a signed permutation $\Psi \in \Pi$ with block structure. The next proposition reveals necessary conditions for identifiability.

**Proposition C.1.** *Let $\mathcal{G}_m$ be an $m$-domain graph with shared latent nodes $\mathcal{L} = [\ell]$, and let $P_X \in \mathcal{M}(\mathcal{G}_m)$ with representation $(B, P)$. Suppose that $m \geq 2$ and that everything except the assumption about pairwise different error distributions in Conditions (C1) and (C2) is satisfied. Then, the joint distribution is not identifiable if one of the following holds:*

*(i) There is $i, j \in \mathcal{L}$ such that $P_i = P_j$ or $P_i = -P_j$.*

*(ii) There is $i \in \mathcal{L}$ and $j \in I_e$ for some $e \in [m]$ such that $P_i = P_j$ or $P_i = -P_j$.*

*(iii) For all $e \in [m]$ there is $i_e \in I_e$ such that $P_{i_e} = P_{j_f}$ or $P_{i_e} = -P_{i_f}$ for all $e \neq f$.*

*Proof.* For each of the three cases, we will construct another $m$-domain graph $\mathcal{G}_m = (\mathcal{H} \cup V, D)$ such that for suitable representations $(\widetilde{B}, \widetilde{P})$ of distributions in $\mathcal{M}(\widetilde{\mathcal{G}}_m)$, the left-hand side of (9) holds, but the right-hand side is violated.

To prove the statement for case (i), let $i, j \in \mathcal{L}$ and assume that $P_i = P_j$. We define the $m$-domain graph $\widetilde{\mathcal{G}}_m = (\widetilde{\mathcal{H}} \cup V, \widetilde{D})$ to be the almost same graph as $\mathcal{G}_m = (\mathcal{H} \cup V, D)$, we only "swap" the roles of the latent nodes $i$ and $j$ on an arbitrary domain $e \in [m]$. That is, for each $v \in V_e$, if there was an edge $i \to v$ in $D$, we remove that edge from $\widetilde{D}$ and add the edge $j \to v$ instead, and vice versa. Otherwise, the graph $\widetilde{\mathcal{G}}_m$ has the same structure as $\mathcal{G}_m$. Now, let $\widetilde{P} = P$ and define a the matrix $\widetilde{B}$ to be the same matrix as $B$, except for the subcolumns $\widetilde{B}_{V_e,i} := B_{V_e,j}$ and $\widetilde{B}_{V_e,j} := B_{V_e,i}$, that is, we swapped $B_{V_e,i}$ and $B_{V_e,j}$. Then the pair $(\widetilde{B}, \widetilde{P})$ is a representation of some distribution in $\mathcal{M}(\widetilde{\mathcal{G}}_m)$. Recall from the proof of Theorem 3.1 that Condition (C2) implies that the matrix $B_{V_e,S_e}$ has full column rank. Since we only swapped columns in $\widetilde{B}_{V_e,\widetilde{S}_e}$, it still has full column rank. Moreover, observe that the left hand side of (9) is satisfied since $P_i = P_j$, that is, the marginal distributions on the single domains coincide.

However, now consider another domain $f \in [m]$ and the submatrices

$$B_{V_e \cup V_f, \{i,j\}} = \begin{pmatrix} B_{V_e,i} & B_{V_e,j} \\ B_{V_f,i} & B_{V_f,j} \end{pmatrix} \quad \text{and} \quad \widetilde{B}_{V_e \cup V_f, \{i,j\}} = \begin{pmatrix} B_{V_e,j} & B_{V_e,i} \\ B_{V_f,i} & B_{V_f,j} \end{pmatrix}.$$

Since all of the four subcolumns are nonzero and neither $B_{V_e,j}$ is equal to $B_{V_e,i}$ nor $B_{V_f,j}$ is equal to $B_{V_f,i}$, there is no signed permutation matrix $\Omega$ such that $B_{V_e \cup V_f, \{i,j\}} = \widetilde{B}_{V_e \cup V_f, \{i,j\}}\Omega$. Hence, there is also no larger signed permutation matrix $\Psi$ such that $B = \widetilde{B}\Psi$. We conclude that the right-hand side of (9) is violated and the joint distribution is not identifiable. Finally, note that the above arguments also hold if $P_i = -P_j$ by adding "$-$" signs in appropriate places.

The proof for case (ii) works with exactly the same construction. That is, for $i \in \mathcal{L}$ and $j \in I_e$ we swap the roles of $i$ and $j$ on the domain $e$. Then, for any other domain $f \in [m]$, we obtain the submatrices

$$B_{V_e \cup V_f, \{i,j\}} = \begin{pmatrix} B_{V_e,i} & B_{V_e,j} \\ B_{V_f,i} & 0 \end{pmatrix} \quad \text{and} \quad \widetilde{B}_{V_e \cup V_f, \{i,j\}} = \begin{pmatrix} B_{V_e,j} & B_{V_e,i} \\ B_{V_f,i} & 0 \end{pmatrix}.$$

By the same arguments as before, this shows that there is no signed permutation matrix $\Psi$ such that $B = \widetilde{B}\Psi$ and, hence, the joint distribution is not identifiable.

To prove case (iii), we consider a slightly different construction. However, we also assume that $P_{i_e} = P_{i_f}$ for all $e \neq f$, since for $P_{i_e} = -P_{i_f}$ we only have to add some "$-$" signs in the following. We define the $m$-domain graph $\widetilde{\mathcal{G}}_m = (\widetilde{\mathcal{H}} \cup V, \widetilde{D})$ by identifying the nodes $i_e, e \in [m]$ with a new

node $k$. That is, $\widetilde{\mathcal{L}} = \mathcal{L} \cup \{k\}$ and $\widetilde{\mathcal{H}} = (\bigcup_{e \in [m]} I_e \setminus \{i_e\}) \cup \widetilde{\mathcal{L}}$. For $i \in \widetilde{\mathcal{H}} \setminus \{k\}$ and $v \in V$, the edge set $\widetilde{D}$ contains an edge $i \to v$ if and only if the edge $i \to v$ is in $D$. For the node $k \in \widetilde{\mathcal{H}}$ and $v \in V$, we put an edge $k \to v$ in $\widetilde{D}$ if and only if there is an edge $i_e \to v$ in $D$ for some $e \in [m]$.

Now, define the matrix $\widetilde{B}$ such that $\widetilde{B}_{V,\widetilde{\mathcal{H}} \setminus \{k\}} := B_{V,\widetilde{\mathcal{H}} \setminus \{k\}}$ and $\widetilde{B}_{V_e, k} := B_{V_e, i_e}$ for all $e \in [m]$. Then the pair $(\widetilde{B}, \widetilde{P})$ is a representation of some distribution in $\mathcal{M}(\widetilde{\mathcal{G}}_m)$. Moreover, each submatrix $\widetilde{B}_{V_e, \widetilde{S}_e}$ is equal to $B_{V_e, S_e}$ up to relabeling of the columns. That is, the column that is labeled by $i_e$ in $B_{V_e, S_e}$ is now labeled by $k$ in $\widetilde{B}_{V_e, \widetilde{S}_e}$. We define the measure $\widetilde{P}$ such that $\widetilde{P}_{\widetilde{\mathcal{H}} \setminus \{k\}} = P_{\widetilde{\mathcal{H}} \setminus \{k\}}$ and $\widetilde{P}_k = P_{i_e}$ for all $e \in [m]$. Then the pair $(\widetilde{B}, \widetilde{P})$ is a representation of some distribution in $\mathcal{M}(\widetilde{\mathcal{G}}_m)$ and, in particular, the left hand side of (9) is satisfied. That is, the marginal distributions coincide on each domain. However, the number of shared latent variables in both $m$-domain graphs is different since we have $\widetilde{\ell} = \ell + 1$. We conclude that the joint distribution is not identifiable. $\square$

The proposition states that it is in most cases necessary that error distributions are pairwise different. However, in two cases the same error distributions still lead to identifiability. First, if $i, j \in I_e$, then the corresponding error distributions may be the same and the joint distribution is still identifiable. Similarly, if there are latent nodes $i_e$ in a few domains $e \in [m]$ such that the corresponding error distributions coincide, but there is at least one domain $f \in [m]$ where there is no latent node with the same error distribution, then the joint distribution is also identifiable. Both can be seen by taking the the proofs of Theorem 3.1 and Proposition C.1 together.

**Gaussian Errors.** Without additional assumptions to those in Section 3, it is impossible to recover the joint distribution if the distributions of the errors $\varepsilon_i$ of the latent structural equation model in Equation (1) are Gaussian. In this case, the distribution of $Z$ as well as the distribution of each observed random vector $X^e$ is determined by the covariance matrix only. The observed covariance matrix in domain $e \in [m]$ is given by $\Sigma^e = G^e \mathrm{Cov}[Z_{\mathcal{L} \cup I_e}](G^e)^\top$. However, knowing $\Sigma^e$ gives no information about $Z_{\mathcal{L} \cup I_e}$ other than $\mathrm{rank}(\Sigma^e) = |\mathcal{L}| + |I_e|$, that is, we cannot distinguish which latent variables are shared and which ones are domain-specific. This is formalized in the following lemma.

**Lemma C.2.** *Let $\Sigma$ be any $d \times d$ symmetric positive semidefinite matrix of rank $p$ and let $\Xi$ be another arbitrary $p \times p$ symmetric positive definite matrix. Then there is $G \in \mathbb{R}^{d \times p}$ such that $\Sigma = G \Xi G^\top$.*

*Proof.* Let $\Sigma$ be a $d \times d$ symmetric positive semidefinite matrix of rank $p$. Then, $\Sigma$ has a decomposition similar to the Cholesky decomposition; see e.g. Gentle (1998, Section 3.2.2). That is, there exists a unique matrix $T$, such that $A = TT^\top$, where $T$ is a lower triangular matrix with $p$ positive diagonal elements and $d - p$ columns containing all zeros. Define $\widetilde{T}$ to be the $d \times p$ matrix containing only the non-zero columns of $T$.

On the other hand, let $\Xi$ be a symmetric positive definite $p \times p$ matrix. By the usual Cholesky decomposition (Lyche, 2020, Section 4.2.1), there exists a unique $p \times p$ lower triangular matrix $L$ with positive diagonal elements such that $\Xi = LL^\top$. Now, define $G := \widetilde{T} L^{-1} \in \mathbb{R}^{d \times p}$. Then,

$$\Sigma = \widetilde{T} \widetilde{T}^\top = \widetilde{T} L^{-1} L L^\top L^{-\top} \widetilde{T}^\top = G \Xi G^\top. \qquad \square$$

Due to Lemma C.2 it is necessary to consider non-Gaussian distributions to obtain identifiability of the joint distribution.

**Example C.3.** In the Gaussian case we cannot distinguish whether the two observed domains in Figure 5 share a latent variable or not. Said differently, the observed marginal distributions may either be generated by the mechanism defined by graph (a) or graph (b) and there is no way to distinguish from observational distributions only.

**Sparsity Assumptions.** Let $B \in \mathrm{Im}(\phi_{\mathcal{G}_m})$ for an $m$-domain graph $\mathcal{G}_m = (\mathcal{H} \cup V, D)$ and suppose that we are given the matrix $B_\mathcal{L} = G_{V,\mathcal{L}}(I - A_{\mathcal{L},\mathcal{L}})^{-1}$, that is, we are given the submatrix with columns indexed by the shared latent nodes. Now, assume that the graph does not impose any sparsity restrictions on $G_{V,\mathcal{L}}$, which means that the set $\mathbb{R}^{D_{V\mathcal{L}}}$ of possible matrices $G_{V,\mathcal{L}}$ is equal to $\mathbb{R}^{|V| \times |\mathcal{L}|}$. Then, the set of possible matrices $B_\mathcal{L}$ is also unrestricted, that is, $B_\mathcal{L}$ can be any matrix in $\mathbb{R}^{|V| \times |\mathcal{L}|}$ no matter the form of the matrix $A_{\mathcal{L},\mathcal{L}} \in \mathbb{R}^{D_{\mathcal{L}\mathcal{L}}}$. In other words, for arbitrary shared latent graphs

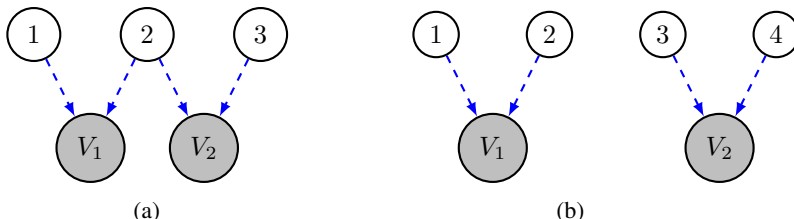

Figure 5: Compact versions of two 2-domain graphs. In both graphs, both domains have two latent parents. In setup (a) there is a shared latent parent while in setup (b) there is not.

$\mathcal{G}_\mathcal{L} = (\mathcal{L}, D_{\mathcal{L}\mathcal{L}})$ and arbitrary corresponding parameter matrices $A_{\mathcal{L},\mathcal{L}} \in \mathbb{R}^{D_{\mathcal{L}\mathcal{L}}}$, we don't get any restrictions on the matrix $B_\mathcal{L}$. Therefore, it is impossible to infer $A_{\mathcal{L},\mathcal{L}}$ from $B_\mathcal{L}$.

Condition (C3) requires that there are two partial pure children for every shared latent node $k \in \mathcal{L}$, which implies that there are $2|\mathcal{L}|$ rows in $G_{V,\mathcal{L}}$ in which only one entry may be nonzero. While we show in Theorem 4.4 that this condition is sufficient for identifiability of $A_{\mathcal{L},\mathcal{L}}$, we leave it open for future work to find a necessary condition.

## D    Algorithms for Finite Samples

We adjust Algorithm 1 such that it is applicable in the empirical data setting. That is, rather than the exact distribution $P_{X^e}$, we have a matrix of observations $\boldsymbol{X}^e \in \mathbb{R}^{d_e \times n_e}$ in each domain $e \in [m]$. The sample size $n_e$ might be different across domains. We denote $n_{\min} = \min_{e \in [m]} n_e$ and $n_{\max} = \max_{e \in [m]} n_e$. For implementing linear ICA on finite samples, multiple well developed algorithms are available, e.g., FastICA (Hyvärinen, 1999; Hyvärinen and Oja, 2000), Kernel ICA (Bach and Jordan, 2003) or JADE (Cardoso and Souloumiac, 1993). Applying them, we obtain a measure $\widehat{P}_i^e$ which is an estimator of the true measure $P_i^e$ in Algorithm 1, Line 4.

The remaining challenge is the matching in Line 6 of Algorithm 1. For finite samples, the distance between empirical distributions is almost surely not zero although the true underlying distributions might be equal. In this section, we provide a matching strategy based on the two-sample Kolmogorov-Smirnov test (van der Vaart and Wellner, 1996, Section 3.7). We match two distributions if they are not significantly different. During this process, there might occur false discoveries, that is, distributions are matched that are actually not the same. We show that the probability of falsely discovering shared nodes shrinks exponentially with the number of domains.

For two univariate Borel probability measures $P_i, P_j$, with corresponding cumulative distribution functions $F_i, F_j$, the Kolmogorov-Smirnov distance is given by the $L^\infty$-distance

$$d_{\text{KS}}(P_i, P_j) = \|F_i - F_j\|_\infty = \sup_{x \in \mathbb{R}} |F_i(x) - F_j(x)|.$$

The two-sample Kolmogorov-Smirnov test statistic for the null hypothesis $H_0 : d_{\text{KS}}(P_i^e, P_j^f) = 0$ is given by

$$T(\widehat{P}_i^e, \widehat{P}_j^f) = \sqrt{\frac{n_e n_f}{n_e + n_f}} d_{\text{KS}}(\widehat{P}_i^e, \widehat{P}_j^f). \tag{10}$$

It is important to note that $\widehat{P}_i^e$ is not an empirical measure in the classical sense since it is not obtained from data sampled directly from the true distribution $P_i^e$. In addition to the sampling error there is the uncertainty of the ICA algorithm. However, in the analysis we present here, we will neglect this error and treat $\widehat{P}_i^e$ as an empirical measure. In this case, under $H_0$, the test statistic $T(\widehat{P}_i^e, \widehat{P}_j^f)$ converges in distribution to $\|B\|_\infty$, where $B$ is a Brownian bridge from 0 to 1 (van der Vaart and Wellner, 1996, Section 2.1). For a given level $\alpha \in (0, 1)$, we choose the critical value as $c_\alpha = \inf\{t : P(\|B\|_\infty > t) \leq \alpha\}$ and reject $H_0$ if $T(\widehat{P}_i^e, \widehat{P}_j^f) > c_\alpha$.

**Definition D.1.** Let $\alpha \in (0, 1)$ and suppose the distributions $\{\widehat{P}_1^e, \ldots, \widehat{P}_{\widehat{s}_e}^e\}$ and $\{\widehat{P}_1^f, \ldots, \widehat{P}_{\widehat{s}_f}^f\}$ are given for two domains $e, f \in [m]$. Define

$$\Omega_\alpha(\widehat{P}_i^e, \widehat{P}_j^f) = \begin{cases} 1 & \text{if } T_{ij}^{ef} \leq c_\alpha \text{ and } T_{ij}^{ef} = \min\{\min_{k \in [\widehat{s}_f]} T_{ik}^{ef}, \min_{k \in [\widehat{s}_e]} T_{kj}^{ef}\}, \\ 0 & \text{else}, \end{cases}$$

where $T_{ij}^{ef} = \min\{T(\widehat{P}_i^e, \widehat{P}_j^f), T(\widehat{P}_i^e, -\widehat{P}_j^f)\}$. We say that $\widehat{P}_i^e, \widehat{P}_j^f$ are *matched* if $\Omega_\alpha(\widehat{P}_i^e, \widehat{P}_j^f) = 1$.

Definition D.1 essentially states that two measures are matched if the test statistic (10) is not significantly large and the null hypothesis cannot be rejected. Taking the minimum of $T(\widehat{P}_i^e, \widehat{P}_j^f)$ and $T(\widehat{P}_i^e, -\widehat{P}_j^f)$ accounts for the sign indeterminacy of linear ICA. For two fixed domains $e, f \in [m]$, if it happens that the statistic $T_{ij}^{ef}$ for multiple pairs $(i, j)$ is small enough, then the pair with the minimal value of the statistic is matched. Note that one may use any other test than the Kolmogorov-Smirnov test to define a matching as in Definition D.1. We discover a shared latent node if it is matched consistently across domains.

**Definition D.2.** Let $C = (i_1, \ldots, i_m)$ be a tuple with $m$ elements such that $i_e \in [\widehat{s}_e]$. Then we say that $C$ determines a *shared node* if $\Omega_\alpha(\widehat{P}_{i_e}^e, \widehat{P}_{i_f}^f) = 1$ for all $i_e, i_f \in C$.

Inferring the existence of a shared node which does not actually exist may be considered a more serious error than inferring a shared node determined by a set $C$, where only some components of $C$ are wrongly matched. In the following theorem we show that the probability of falsely discovering shared nodes shrinks exponentially with the number of wrongly matched components.

**Theorem D.3.** *Let $C = (i_1, \ldots, i_m)$ be a tuple with $m$ elements such that $i_e \in [\widehat{s}_e]$. Let $g : \mathbb{N} \times \mathbb{R}_{\geq 0} \to \mathbb{R}_{\geq 0}$ be a function that is monotonically decreasing in $n \in \mathbb{N}$ and assume the following:*

*(i)* $P(d_{KS}(\widehat{P}_{i_e}^e, P_{i_e}^e) > x) \leq g(n_e, x)$ *for all $e \in [m]$ and for all $x \geq 0$.*

*(ii)* *There is $E \subseteq [m]$ with $|E| \geq 2$ and a constant $\kappa > 0$ such that $d_{KS}(P_{i_e}^e, P_{i_f}^f) \geq \kappa$ and $d_{KS}(P_{i_e}^e, -P_{i_f}^f) \geq \kappa$ for all $e \in E$, $f \in [m]$ with $e \neq f$.*

*Then*

$$P(C \text{ determines a shared node}) \leq g\left(n_{\min}, \max\left\{\frac{\kappa}{2} - \frac{\sqrt{n_{\max}}}{\sqrt{2}\, n_{\min}} c_\alpha, 0\right\}\right)^{|E|-1}.$$

*Proof of Theorem D.3.* Let $C = (i_1, \ldots, i_m)$ and $g$ be as in the statement of the theorem. W.l.o.g. we assume that $E = \{1, \ldots, |E|\}$ and that $T(\widehat{P}_{i_e}^e, \widehat{P}_{i_f}^f) \leq T(\widehat{P}_{i_e}^e, -\widehat{P}_{i_f}^f)$. Observe that $C$ determines a shared node if and only if

$$\sum_{e<f} \Omega_\alpha(\widehat{P}_{i_e}^e, \widehat{P}_{i_f}^f) = \binom{m}{2}.$$

Now, we have

$$P\left(\sum_{e<f} \Omega_\alpha(\widehat{P}_{i_e}^e, \widehat{P}_{i_f}^f) = \binom{m}{2}\right) = P\left(\bigcap_{e<f}\left\{\Omega_\alpha(\widehat{P}_{i_e}^e, \widehat{P}_{i_f}^f) = 1\right\}\right)$$

$$\leq P\left(\bigcap_{e<f}\left\{T_{ij}^{ef} \leq c_\alpha\right\}\right)$$

$$= P\left(\bigcap_{e<f}\left\{T(\widehat{P}_{i_e}^e, \widehat{P}_{i_f}^f) \leq c_\alpha\right\} \cup \left\{T(\widehat{P}_{i_e}^e, -\widehat{P}_{i_f}^f) \leq c_\alpha\right\}\right)$$

$$= P\left(\bigcap_{e<f}\left\{T(\widehat{P}_{i_e}^e, \widehat{P}_{i_f}^f) \leq c_\alpha\right\}\right)$$

$$\leq P\left(\bigcap_{\substack{e \in E, f \in [m] \\ e<f}}\left\{T(\widehat{P}_{i_e}^e, \widehat{P}_{i_f}^f) \leq c_\alpha\right\}\right). \tag{11}$$

By the triangle inequality we have

$$d_{KS}(P_{i_e}^e, P_{i_f}^f) \leq d_{KS}(P_{i_e}^e, \widehat{P}_{i_e}^e) + d_{KS}(\widehat{P}_{i_e}^e, \widehat{P}_{i_f}^f) + d_{KS}(\widehat{P}_{i_f}^f, P_{i_f}^f). \tag{12}$$

Moreover, if $e \in E$ and $f \in [m]$, then we have by Condition (ii)

$$d_{\text{KS}}(P^e_{i_e}, P^f_{i_f}) \geq \kappa. \tag{13}$$

Using (12) and (13) together with the fact $\sqrt{\frac{n_e n_f}{n_e + n_f}} \geq \frac{n_{\min}}{\sqrt{2}n_{\max}}$, we obtain the following chain of implications for $e \in E$ and $f \in [m]$:

$$T(\widehat{P}^e_{i_e}, \widehat{P}^f_{i_f}) \leq c_\alpha$$

$$\Longleftrightarrow \sqrt{\frac{n_e n_f}{n_e + n_f}} d_{\text{KS}}(\widehat{P}^e_{i_e}, \widehat{P}^f_{i_f}) \leq c_\alpha$$

$$\Longrightarrow \frac{n_{\min}}{\sqrt{2}n_{\max}} \left( d_{\text{KS}}(P^e_{i_e}, P^f_{i_f}) - d_{\text{KS}}(\widehat{P}^e_{i_e}, P^e_{i_e}) - d_{\text{KS}}(\widehat{P}^f_{i_f}, P^f_{i_f}) \right) \leq c_\alpha$$

$$\Longrightarrow \frac{n_{\min}}{\sqrt{2}n_{\max}} \left( \kappa - d_{\text{KS}}(\widehat{P}^e_{i_e}, P^e_{i_e}) - d_{\text{KS}}(\widehat{P}^f_{i_f}, P^f_{i_f}) \right) \leq c_\alpha$$

$$\Longleftrightarrow d_{\text{KS}}(\widehat{P}^e_{i_e}, P^e_{i_e}) + d_{\text{KS}}(\widehat{P}^f_{i_f}, P^f_{i_f}) \geq \kappa - \frac{\sqrt{2}n_{\max}}{n_{\min}} c_\alpha$$

$$\Longrightarrow \left\{ d_{\text{KS}}(\widehat{P}^e_{i_e}, P^e_{i_e}) \geq \frac{\kappa}{2} - \frac{\sqrt{n_{\max}}}{\sqrt{2}\,n_{\min}} c_\alpha \right\} \text{ or } \left\{ d_{\text{KS}}(\widehat{P}^f_{i_f}, P^f_{i_f}) \geq \frac{\kappa}{2} - \frac{\sqrt{n_{\max}}}{\sqrt{2}\,n_{\min}} c_\alpha \right\}.$$

Now, consider the event $\bigcap_{\substack{e \in E, f \in [m] \\ e < f}} \{ T(\widehat{P}^e_{i_e}, \widehat{P}^f_{i_f}) \leq c_\alpha \}$. On this event, there cannot be two elements $e, f \in E$ such that both

$$\left\{ d_{\text{KS}}(\widehat{P}^e_{i_e}, P^e_{i_e}) < \frac{\kappa}{2} - \frac{\sqrt{n_{\max}}}{\sqrt{2}\,n_{\min}} c_\alpha \right\} \text{ and } \left\{ d_{\text{KS}}(\widehat{P}^f_{i_f}, P^f_{i_f}) < \frac{\kappa}{2} - \frac{\sqrt{n_{\max}}}{\sqrt{2}\,n_{\min}} c_\alpha \right\}.$$

To see this recall that $E \subseteq [m]$. We conclude that it must hold $\{ d_{\text{KS}}(\widehat{P}^e_{i_e}, P^e_{i_e}) \geq \frac{\kappa}{2} - \frac{\sqrt{n_{\max}}}{\sqrt{2}\,n_{\min}} c_\alpha \}$ for all but at most one element of $E$. We denote this exceptional element by $e^* \in E$. Taking up (11), we get the following:

$$P \left( \sum_{e < f} \Omega_\alpha(\widehat{P}^e_{i_e}, \widehat{P}^f_{i_f}) = \binom{m}{2} \right)$$

$$\leq P \left( \bigcap_{\substack{e \in E, f \in [m] \\ e < f}} \left\{ T(\widehat{P}^e_{i_e}, \widehat{P}^f_{i_f}) \leq c_\alpha \right\} \right)$$

$$\leq P \left( \bigcap_{e \in E \setminus \{e^*\}} \left\{ d_{\text{KS}}(\widehat{P}^e_{i_e}, P^e_{i_e}) \geq \frac{\kappa}{2} - \frac{\sqrt{n_{\max}}}{\sqrt{2}\,n_{\min}} c_\alpha \right\} \right)$$

$$= \prod_{e \in E \setminus \{e^*\}} P \left( d_{\text{KS}}(\widehat{P}^e_{i_e}, P^e_{i_e}) \geq \frac{\kappa}{2} - \frac{\sqrt{n_{\max}}}{\sqrt{2}\,n_{\min}} c_\alpha \right) \tag{14}$$

$$= \prod_{e \in E \setminus \{e^*\}} P \left( d_{\text{KS}}(\widehat{P}^e_{i_e}, P^e_{i_e}) \geq \max \left\{ \frac{\kappa}{2} - \frac{\sqrt{n_{\max}}}{\sqrt{2}\,n_{\min}} c_\alpha, 0 \right\} \right) \tag{15}$$

$$\leq g \left( n_{\min}, \max \left\{ \frac{\kappa}{2} - \frac{\sqrt{n_{\max}}}{\sqrt{2}\,n_{\min}} c_\alpha, 0 \right\} \right)^{|E|-1}. \tag{16}$$

The last three steps need more explanation: Equality (14) follows from the fact that domains are unpaired. That is, the distances $d_{\text{KS}}(\widehat{P}^e_{i_e}, P^e_{i_e})$ and $d_{\text{KS}}(\widehat{P}^f_{i_f}, P^f_{i_f})$ are pairwise independent for different domains $e, f \in [m]$. Equality (15) is trivial since $d_{\text{KS}}(\widehat{P}^e_{i_e}, P^e_{i_e}) \geq 0$. Finally, Inequality (16) follows from Condition (i) and that the function $g$ is monotonically decreasing in $n$. We also used the fact that $|E \setminus \{e^*\}| = |E| - 1$. $\qquad\square$

If $\widehat{P}^e_i$ were an empirical measure in the classical sense, then Condition (i) in Theorem D.3 translates to the well-known Dvoretzky–Kiefer–Wolfowitz inequality, that is, the function $g$ is given by $g(n, x) =$

**Algorithm 3** IdentifyJointDistributionEmpirical
___

1: **Hyperparameters:** $\gamma > 0$, $\alpha \in (0,1)$.
2: **Input:** Matrix of observations $\boldsymbol{X}^e \in \mathbb{R}^{d_e \times n_e}$ for all $e \in [m]$.
3: **Output:** Number of shared latent variables $\widehat{\ell}$, matrix $\widehat{B}$ and probability measure $\widehat{P}$.
4: **for** $e \in [m]$ **do**
5:   Linear ICA: Use any linear ICA algorithm to obtain a mixing matrix $\widetilde{B}^e \in \mathbb{R}^{d_e \times \widehat{s}_e}$, where $\widehat{s}_e = \mathrm{rank}_\gamma(\boldsymbol{X}^e(\boldsymbol{X}^e)^\top)$. Compute the matrix $\tilde{\boldsymbol{\eta}}^e = (\widetilde{B}^e)^\dagger \boldsymbol{X}^e \in \mathbb{R}^{\widehat{s}_e \times n_e}$, where $(\widetilde{B}^e)^\dagger$ is the Moore-Penrose pseudoinverse of $\widetilde{B}^e$.
6:   Scaling: Let $\Delta^e$ be a $\widehat{s}_e \times \widehat{s}_e$ diagonal matrix with entries $\Delta_{ii}^e = \frac{1}{n_e}[\tilde{\boldsymbol{\eta}}^e(\tilde{\boldsymbol{\eta}}^e)^\top]_{ii}$. Define $\widehat{B}^e = \widetilde{B}^e(\Delta^e)^{-1/2}$ and $\boldsymbol{\eta}^e = (\Delta^e)^{-1/2}\tilde{\boldsymbol{\eta}}^e$.
7:   Let $\widehat{P}^e$ be the estimated probability measure with independent marginals such that $\widehat{P}_i^e$ is the empirical measure of the row $\boldsymbol{\eta}_{i,*}^e$.
8: **end for**
9: Matching: Let $\widehat{\ell}$ be the maximal number such that there is a signed permutation matrix $Q^e$ in each domain $e \in [m]$ such that

$$\Omega_{\alpha_t}([(Q^e)^\top \# \widehat{P}^e]_i, [(Q^f)^\top \# \widehat{P}^f]_i) = 1$$

   for all $i = 1, \ldots, \widehat{\ell}$ and all $f \neq e$, where $\alpha_t = \alpha/t$ with $t = 2\sum_{e<f} \widehat{s}_e \widehat{s}_f$. Let $\widehat{\mathcal{L}} = \{1, \ldots, \widehat{\ell}\}$.

10: Construct the matrix $\widehat{B}$ and the tuple of probability measures $\widehat{P}$ given by

$$\widehat{B} = \begin{pmatrix} [\widehat{B}^1 Q^1]_{\widehat{\mathcal{L}}} & [\widehat{B}^1 Q^1]_{[\widehat{s}_1]\setminus\widehat{\mathcal{L}}} & & \\ \vdots & & \ddots & \\ [\widehat{B}^m Q^m]_{\widehat{\mathcal{L}}} & & & [\widehat{B}^m Q^m]_{[\widehat{s}_m]\setminus\widehat{\mathcal{L}}} \end{pmatrix} \quad \text{and} \quad \widehat{P} = \begin{pmatrix} [(Q^1)^\top \# \widehat{P}^1]_{\widehat{\mathcal{L}}} \\ [(Q^1)^\top \# \widehat{P}^1]_{[\widehat{s}_1]\setminus\widehat{\mathcal{L}}} \\ \vdots \\ [(Q^m)^\top \# \widehat{P}^m]_{[\widehat{s}_m]\setminus\widehat{\mathcal{L}}} \end{pmatrix}.$$

11: **return** $(\widehat{\ell}, \widehat{B}, \widehat{P})$.
___

**Algorithm 4** IdentifySharedGraphEmpirical
___

1: **Hyperparameters:** $\gamma > 0$.
2: **Input:** Matrix $B^* \in \mathbb{R}^{|V| \times \ell}$.
3: **Output:** Parameter matrix $\widehat{A} \in \mathbb{R}^{\ell \times \ell}$.
4: Remove rows $B_{i,\mathcal{L}}^*$ with $\|B_{i,\mathcal{L}}^*\|_2 \leq \gamma$ from the matrix $B^*$.
5: Find tuples $(i_k, j_k)_{k \in \mathcal{L}}$ with the smallest possible scores $\sigma_{\min}(B_{\{i_k,j_k\},\mathcal{L}}^*)$ such that

   (i) $i_k \neq j_k$ for all $k \in \mathcal{L}$ and $\{i_k, j_k\} \cap \{i_q, j_q\} = \emptyset$ for all $k, q \in \mathcal{L}$ such that $k \neq q$ and
   (ii) $\sigma_{\min}(B_{\{i_k,i_q\},\mathcal{L}}^*)| > \gamma$ for all $k, q \in \mathcal{L}$ such that $k \neq q$.

6: Let $I = \{i_1, \ldots, i_\ell\}$ and consider the matrix $B_{I,\mathcal{L}}^* \in \mathbb{R}^{\ell \times \ell}$.
7: Find two permutation matrices $R_1$ and $R_2$ such that $W = R_1 B_{I,\mathcal{L}}^* R_2$ is as close as possible to lower triangular. This can be measured, for example, by using $\sum_{i<j} W_{ij}^2$.
8: Multiply each column of $W$ by the sign of its corresponding diagonal element. This yields a new matrix $\widetilde{W}$ with all diagonal elements positive.
9: Divide each row of $\widetilde{W}$ by its corresponding diagonal element. This yields a new matrix $\widetilde{W}'$ with all diagonal elements equal to one.
10: Compute $\widehat{A} = I - (\widetilde{W}')^{-1}$.
11: **return** $\widehat{A}$.
___

$2\exp(-2nx^2)$. Given a tuple $C = (i_1, \ldots, i_m)$ that defines a shared node, Condition (ii) is an assumption on the number of wrongly matched components. The most extreme case is when the shared node does not actually exist and all components are wrongly matched. That is, the measures $\widehat{P}^e_{i_e}$ and $\widehat{P}^f_{i_f}$ are matched even though $d_{\text{KS}}(P^e_{i_e}, P^f_{i_f}) \neq 0$ and $d_{\text{KS}}(P^e_{i_e}, -P^f_{i_f}) \neq 0$ *for all* $e, f \in [m]$. On the other hand, if $|E| \ll m$, then $C$ determines a shared node where the majority of the components are correctly matched.

If $g(n, x) \to 0$ for $n \to \infty$ and $x > 0$, the statement of the theorem becomes meaningful under the constraint $\sqrt{n_{\max}}/n_{\min} \to 0$. In this case, the probability that a given tuple $C$ with wrong components $E$ determines a shared node goes to zero for large sample sizes $n_{\min}$. As noted, the probability of falsely discovering a shared node decreases exponentially with the number of wrongly matched components $|E|$. In the extreme case, this means that the probability of falsely discovering shared nodes with all components wrongly matched, i.e., $E = [m]$, decreases exponentially with the number of domains $m$.

Theorem D.3 also tells us that the probability of falsely matching two measures $\widehat{P}^e_i$ and $\widehat{P}^f_j$ becomes zero if the sample size grows to infinity and the linear ICA algorithm is consistent. However, with finite samples we might fail to match two measures where the underlying true measures are actually the same, i.e., we falsely reject the true null hypothesis $H_0$. Thus, we might be overly conservative in detecting shared nodes due to a high family-wise error rate caused by multiple testing. We suggest to correct the level $\alpha$ to account for the amount of tests carried out. One possibility is to apply a Bonferroni-type correction. The total number of tests is given by $t = 2\sum_{e<f} \widehat{s}_e \widehat{s}_f$. This means that an adjusted level is given by $\alpha_t = \alpha/t$ and instead of the critical value $c_\alpha$ we consider the adjusted critical value $c_{\alpha_t}$.

Algorithm 3 is the finite sample version of Algorithm 1 with the matching $\Omega_\alpha$ defined in Definition D.1. To determine the number of independent components for the linear ICA step in each domain, we need to check the rank$(\boldsymbol{X}^e(\boldsymbol{X}^e)^\top)$. We specify the rank of a matrix $M$ as number of singular values which are larger than a certain threshold $\gamma$ and denote it by rank$_\gamma(M)$.

In Algorithm 4 we also provide a finite sample version of Algorithm 2 where we only have the approximation $B^\star = \widehat{B}_{\mathcal{L}} \approx B_{\mathcal{L}}\Psi_{\mathcal{L}}$ for a signed permutation matrix $\Psi_{\mathcal{L}}$. For a matrix $M$, we denote by $\sigma_{\min}(M)$ the smallest singular value.

# E  Error Distributions in Simulations

We specify $\mathcal{L} = \{1, 2, 3\}$, $I_1 = \{4, 5\}$, $I_2 = \{6, 7\}$ and $I_3 = \{8, 9\}$. Note that the set $I_3$ does not exist if the number of domains is $m = 2$. The error distributions in all simulations are specified as follows if not stated otherwise.

$\mathcal{L}$: $\varepsilon_1 \sim \overline{\text{Beta}}(2, 3)$, $\varepsilon_2 \sim \overline{\text{Beta}}(2, 5)$, $\varepsilon_3 \sim \overline{\chi}^2_4$,

$I_1$: $\varepsilon_4 \sim \overline{\text{Gumbel}}(0, 1)$, $\varepsilon_5 \sim \overline{\text{LogNormal}}(0, 1)$,

$I_2$: $\varepsilon_6 \sim \overline{\text{Weibull}}(1, 2)$, $\varepsilon_7 \sim \overline{\text{Exp}}(0.1)$,

$I_3$: $\varepsilon_8 \sim \overline{\text{SkewNormal}}(6)$, $\varepsilon_9 \sim \overline{\text{SkewNormal}}(12)$,

where the overline means that each distribution is standardized to have mean $0$ and variance $1$. Figure 6 shows histograms of the empirical distributions.

# F  Additional Simulation Results

In this section, we make additional experiments. First, we consider another setup where all our assumptions are satisfied but we have more domains and more shared latent variables. Then, we also consider two setups where some of our assumptions are not satisfied.

**Different Setup.** We make additional experiments on a similar scale as in Section 5, but with more shared nodes and less domain specific nodes. This time, we consider $\ell = 5$ shared latent nodes and $|I_e| = 1$ domain-specific latent node in each domain. Moreover, we also consider $m = 4$ domains. The dimensions are given by $d_e = d/m$ for all $e \in [m]$ and $d = 48$. The graphs and edge weights

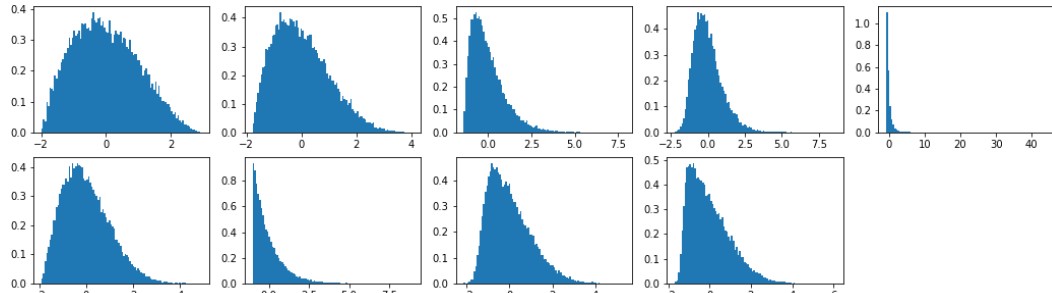

Figure 6: Histograms showing the frequency of 25000 values sampled from the random variables $\varepsilon_i$ with distribution as specified in Appendix E. Each distribution has mean zero and variance one. The first row shows the empirical distributions from $\varepsilon_1$ to $\varepsilon_5$ and the second row from $\varepsilon_6$ to $\varepsilon_9$.

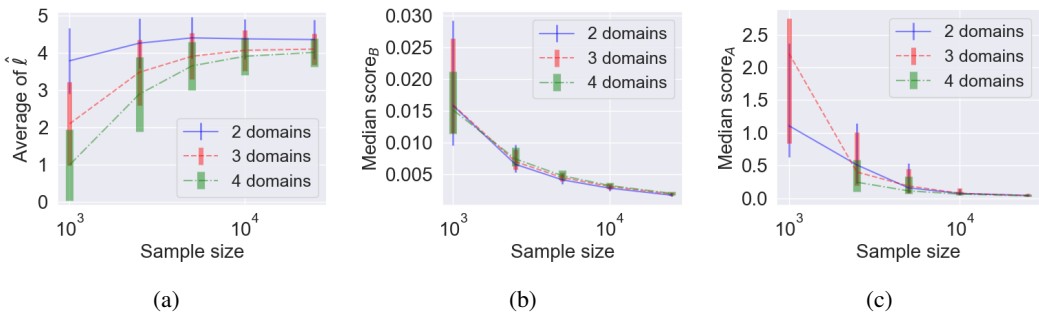

Figure 7: **Simulation results for $\ell = 5$ shared latent nodes.** Logarithmic scale on the $x$-axis. Error bars in (a) are one standard deviation of the mean and in (b) and (c) they are the interquartile range.

are sampled equivalently as in Section 5 in the main paper. We also consider the same distribution of the error variables as specified in Appendix E, where we specify $\mathcal{L} = \{1, 2, 3, 4, 5\}$, $I_1 = \{6\}$, $I_2 = \{7\}$, $I_3 = \{8\}$ and $I_4 = \{9\}$.

Figure 7 shows the results where the scores are equivalent as in the main paper. Once again, we see that the estimation error for the matrices $B_{\mathcal{L}}$ and $A_{\mathcal{L},\mathcal{L}}$ decreases with increasing sample size. This supports our proof of concept and shows that the adapted algorithms are consistent for recovering $B_{\mathcal{L}}$ and $A_{\mathcal{L},\mathcal{L}}$ from finite samples.

**Violated Assumptions.** We consider the same setup as in the main paper in Section 5 with $l = 3$ shared latent nodes, but we fix the number of domains to $m = 3$. In this experiment, we compare the results where data was generated such that all our assumptions are satisfied with two setups where we violate some of the assumptions. In the first setup, we violate Condition (C1) that requires pairwise different error distributions. We specify the error distributions as follows.

$$\mathcal{L}: \ \varepsilon_1 \sim \overline{\text{Beta}}(2, 3), \ \varepsilon_2 \sim \overline{\text{Beta}}(2, 5), \ \varepsilon_3 \sim \overline{\chi}_4^2,$$
$$I_1: \ \varepsilon_4 \sim \overline{\text{Beta}}(2, 3), \ \varepsilon_5 \sim \overline{\text{LogNormal}}(0, 1),$$
$$I_2: \ \varepsilon_6 \sim \overline{\text{Beta}}(2, 5), \ \varepsilon_7 \sim \overline{\text{LogNormal}}(0, 1),$$
$$I_3: \ \varepsilon_8 \sim \overline{\chi}_4^2, \ \varepsilon_9 \sim \overline{\text{LogNormal}}(0, 1),$$

where, as before, the overline means that each distribution is standardized to have mean 0 and variance 1. In the second setup, we do not change the error distributions but we violate Condition (C3) that requires two partial pure children per shared latent node. In this experiment, we do not make any sparsity assumptions on the mixing matrix $G_{V,\mathcal{L}}$.

Figure 8 shows the results of our experiments. As expected, we see in (a) and (b) that identifyability of the joint distribution fails if we do not require pairwise different error distributions. Recovering the joint distribution still works well in the second setup where we violate the partial pure children

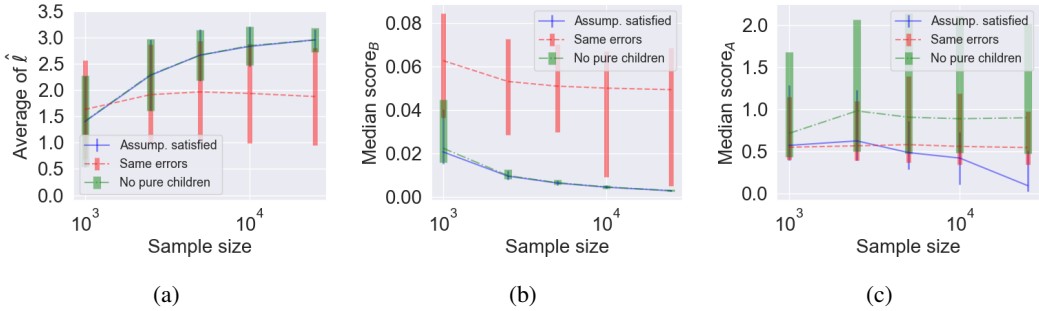

(a)            (b)            (c)

Figure 8: **Simulation results where assumptions are not satisfied.** Logarithmic scale on the $x$-axis. Error bars in (a) are one standard deviation of the mean and in (b) and (c) they are the interquartile range.

conditions. However, identifying the shared latent graph is impossible in this setup, as we explained in Appendix C. This is supported by the experimental results displayed in Figure 8 (c), where we can see that recovery of the shared latent graph does not work when the partial pure children assumption is not satisfied.

# G   Future Work

We see many directions for future work, which include the following.

- Our work and algorithms rely on linear ICA. It would be interesting to study a more direct approach to recover the joint distribution and the causal graph. This might potentially be done by testing certain constraints implied by the model similar as the developments in the LiNGAM literature; see e.g. Shimizu et al. (2011) and Wang and Drton (2020).

- Our results require non-Gaussianity and that both the latent structural equation model and the mixing functions are linear. We consider the linear setup as a basis for any subsequent study of nonlinear cases. For example, recent advances in non-linear ICA allow identifiability of up to linear transformations, see e.g. Khemakhem et al. (2020), Buchholz et al. (2022) and Roeder et al. (2021). Identifiability of a causal representation might then be obtained from identifiability results for the linear model.

- Our sufficient condition for identifiability of the shared latent graph requires two partial pure children per shared latent node. In this regard, it would also be interesting to study necessary conditions; c.f. our discussion in Appendix C.

- This work focused purely on the observational case. However, considering interventional data can be expected to permit relaxing some conditions in both of the key steps, i.e., recovering the joint distribution and the shared latent graph. For example, recent work shows that interventional data allow for identification of the latent graph without sparsity constraints in a single-domain setup (Squires et al., 2023; Ahuja et al., 2023). Extending this to the multi-domain setup is an interesting problem for future work.

- It would be interesting to study the statistical properties of our setup such as theoretical bounds on the accuracy of recovering the matrices $B$ and $A_{\mathcal{L},\mathcal{L}}$ as well as developing algorithms that meet these bounds. Currently, our adapted algorithms for finite samples determine the rank of a matrix by using a threshold for singular values. The algorithms depend on the choice of this parameter and it would be worth studying optimal choices. Moreover, one might consider different methods for determining the rank of a matrix.

- There might be different matching strategies of the estimated error distributions in the finite sample setting. For example, instead of matching pairwise consistently across all domains as we propose in Appendix D, one might find an optimal matching by solving a linear program.

