# OpenReview forum: "Unpaired Multi-Domain Causal Representation Learning"
_NeurIPS.cc/2023/Conference — NeurIPS 2023 spotlight_

### Official Review · Reviewer_AWor · 2023-07-01

**Soundness:** 4 excellent
**Presentation:** 3 good
**Contribution:** 4 excellent
**Rating:** 7
**Confidence:** 3

**Summary:**

The authors study the setting of unsupervised learning where observations belong to several domains, and we only observe the marginal distribution of each domain. A set of latent variables generates the observations, where a subset of latents are shared across domains. The authors provide the first identification results in this setting, assuming that the latents follow a linear SCM, and the observations are an injective linear transformation of the latents. With this model, identifying the (unobserved) joint distribution of the observations equates to identifying the latents. The mapping between the exogenous noises and the observations, as well as the distributions over the exogenous noises, are identified up to signed block permutation. With additional conditions, the authors also identify the causal graph for the latents up to a signed permutation consistent with the topological ordering of the latents. The authors validate their claims with a synthetic data experiment.

**Strengths:**

This is a strong paper that provides the first identifiability results on multiple-domain unsupervised learning where the joint distribution of the observed variables is unobserved. These results are impactful, since this problem setup is well-studied and has practical applications in single-cell biology. Existing approaches are significantly limited by their lack of identifiability, so this paper makes a valuable contribution.

The writing style is rigorous, and definitions, assumptions, and results are explained precisely.

**Weaknesses:**

This paper could be improved with more context on how they are extending existing identification results to achieve theirs. Currently, the authors mention which existing results are being used, but do not provide an intuitive description on why they need to be extended, and how they do so.

The notation could be improved. Capital letters are used to denote matrices, probability measures, vector- and scalar-valued random variables, sets of nodes, and sets of edges. It would improve readability if you used font styles (e.g. lower-case bold for vectors) to differentiate them.

**Questions:**

This paper makes two contributions. The first is to extend identifiable single-domain linear ICA to the multi-domain setting. The second is to extend single-domain graph identifiability to the multi-domain setting. In both cases, can you intuitively describe what about the multi-domain setting prevents the existing results from holding automatically?

For example, if there are no shared latents, the joint can be identified by trivially applying linear ICA separately to each domain. What is the difficulty imposed by a subset of latents being shared, and how is this circumvented?

**Limitations:**

The authors made it explicit that this work is primarily about identifiability, and less about scalable algorithms and evaluation on realistic datasets.

---

> ### Author Rebuttal · Authors · 2023-08-04
>
> We thank the reviewer for the supportive review and helpful comments.
>
> *This paper makes two contributions. The first is to extend identifiable single-domain linear ICA to the multi-domain setting. The second is to extend single-domain graph identifiability to the multi-domain setting. In both cases, can you intuitively describe what about the multi-domain setting prevents the existing results from holding automatically?* \
> *For example, if there are no shared latents, the joint can be identified by trivially applying linear ICA separately to each domain. What is the difficulty imposed by a subset of latents being shared, and how is this circumvented?*
>
> Many thanks for raising this question. As you correctly mention, we extend the identifiability of linear ICA in one domain to obtain identifiability of the joint distribution. Moreover, it is true that the joint can be identified by trivially applying linear ICA separately to each domain when there are no shared latents. This corresponds to the case where the observations of different domains are independent. However, if they are dependent, then the shared latent variables determine the dependency structure of the domains. Thus, it is crucial to determine which of the latent variables are shared and which ones are domain-specific. Assuming non-symmetric, pairwise different error distributions is a sufficient condition to "match" and identify shared distributions. As we discuss in detail in our response to aPgx, this condition is also necessary.
>
> Applying recent results on single-domain latent graph identifiability yields a similar challenge: We recover a latent graph in each domain, but how to integrate them? That is, how to determine which latent factors are shared across domains since they represent conceptually the same? We circumvent this problem by first recovering the joint distribution by extending linear ICA (as described above) and then recover the shared graph from the recovered *large mixing matrix* $\widehat{B}$. Our conditions on sparsity, i.e., two partial pure children, are inspired by the results on single-domain graph identifiability that require pure children. However, we are able to relax the conditions since we use a different proof technique tailored to our setup, i.e., we exploit rank constraints in the *large mixing matrix* that we already identified. This allows us to show that two *partial* pure children *across* domains are sufficient, i.e., a total of 2 across *all* domains. In contrast, single-domain results would require 2 pure children in *every* domain, in addition to the fact that integrating each domain's different graph remains challenging.
>
> In the camera-ready version, we are happy to add this intuition in the introduction in the paragraph where we talk about proof techniques.
>
> *The notation could be improved. Capital letters are used to denote matrices, probability measures, vector- and scalar-valued random variables, sets of nodes, and sets of edges. It would improve readability if you used font styles (e.g. lower-case bold for vectors) to differentiate them.*
>
> Thank you for suggesting a better notation. This is helpful feedback, and we will make the suggested changes in the camera-ready version.

---

> > ### Comment · Reviewer_AWor · 2023-08-17
> >
> > Thank you for your response, and for providing additional intuition regarding your contributions. I think the assumption of pairwise different error distributions is reasonable. I think this is a strong paper, and maintain my score supporting its acceptance.

---

### Official Review · Reviewer_ny5v · 2023-07-03

**Soundness:** 3 good
**Presentation:** 3 good
**Contribution:** 2 fair
**Rating:** 6
**Confidence:** 3

**Summary:**

In this paper, the authors address unpaired multi-domain causal representation. In detail, the authors learn the representations of the observed data from different domains that consist of causally. To achieve this, the authors consider the data generation process where the relationship between the latent variables is linear. Based on this generation process, they prove that the joint distribution of observed data and the shared causal structures of latent variables are identifiable.

**Strengths:**

The authors investigate the causal discovery with latent variables from different domains.

**Weaknesses:**

1.	There are several works about causal discovery with latent variables under linear and multi-domain case like [1], which also considers the shared causal structure among latent variables and provide identification guarantees. It is suggested that the authors should discuss these works.
2.	Moreover, the authors discuss several works about domain translation between unpaired data and claim that none of these works have rigorous identifiability guarantees. However, Multi-domain image generation, image translation, and domain adaptation belong to the proposed setting, and [2][3] have addressed the multi-domain causal representation learning problem recently. And the authors do not consider these works. It is noted that [2][3] considers the multi-domain causal representation learning with nonlinear transformation, which seems to be more general than the proposed setting.
3.	As for the identification of joint distributions, it is not clear why the identification of $l, B$, and $P$ can identify the joint distributions of observed data from multi-domains.
4.	In section 3, the authors assume that the distribution of errors is non-Gaussian for the identification of linear ICA by not allowing asymmetric distribution. But some distribution like the Laplace distribution is symmetric and they can also satisfy the identification of linear ICA.
5.	According to this paper, the authors consider the structure of latent variables to be linear but flexible. In the simulation experiment, the authors only consider three shared latent variables, it is suggested that the authors should consider more latent variables and different structures.
6.	Besides, it is suggested that the authors should consider more compared methods and employ other metrics like recall, and precisions to evaluate the performance of causal discovery.

[1] Causal Discovery with Multi-Domain LiNGAM for Latent Factors
[2] Multi-domain image generation and translation with identifiability guarantees
[3] partial disentanglement for domain adaptation

**Questions:**

Please refer weaknesses

**Limitations:**

Please refer weaknesses

---

> ### Author Rebuttal · Authors · 2023-08-03
>
> We thank the reviewer for their thorough review, which raises several helpful points. We hope that our response addresses their remaining concerns.
>
> # Response to weaknesses
>
> **1.)** We thank the reviewer for pointing out reference [1], which is indeed related to our work, and we will gladly add and discuss this additional reference. In short, the authors of [1] also assume a linear data-generating process on multiple-domains $X=(X_1, \ldots, X_m)$, where
>
> $X = GZ$ and $Z=AZ+\varepsilon$,
>
> but there are important differences. First, $G$ and $A$ are assumed to have block-diagonal structure. Additionally, the authors assume that $Z=H\widetilde{Z}$ for $H \in \mathbb{R}^{|\mathcal{H}|} \times \tilde{q}$ such that $H$ only has one non-zero entry in each row. This allows to rewrite the data-generating process as
>
> $X = GH\widetilde{Z}=\widetilde{G}\widetilde{Z}$ and $\widetilde{Z} = H^{\dagger}AH\widetilde{Z} + H^{\dagger}\varepsilon = \widetilde{B} \widetilde{Z} + \widetilde{\varepsilon},$
>
> where $H^{\dagger}$ denotes the pseudo-inverse.
>
> The authors are then able to prove that  $\widetilde{B}$ and $G$ can be identified. However, this is different from the goal we pursue here. Indeed, the authors do not claim that the matrix $H$ would be identifiable in their Theorem 2. To our understanding, $\widetilde{B}$ has a different interpretation for every possible $H$ in terms of learning the causal relations among the latent variables in $Z$. This said, the authors do consider a setting in Theorem 3, where the number of variables increases ($p \to \infty$), and consistent estimation of $H$ is possible. However, this differs from our setting, where we consider a general but fixed number of observed variables.
>
> At a more technical level, the conditions in [1]  require two pure children, while we prove identifiability under the weaker assumption of two partial pure children.
>
> As noted, we will gladly add [1] to our references and discuss it.
>
> **2.)** Thanks for pointing out the related work on identifiability of the joint distribution. We agree that we should be more careful about our claims on identifiability in prior works and are happy to include the references. However, even though the related work gives identifiability guarantees of the joint distribution in a non-linear setup, it does not tackle identifiability of a *causal* representation.
>
> **3.)** This is clarified in the paragraph after Theorem 3.1: Since we recover $\widehat{B}=B \Psi$ and $\widehat{P} = \Psi^{\top}$#$P$, we have that
>
> $\widehat{B}$#$\widehat{P}$ = $B \Psi$#($\Psi^{\top}$#$P$) =  $B \Psi \Psi^{\top}$#$P$ = $B$#$P$,
>
> where $M$#$P$ denotes the push-forward measure under the linear map $M$. Now, by Definition 2.3, the joint distribution $P_X$ is equal to the push-forward distribution $B$#$P$, which shows that the identification of $B$ and $P$ up to signed permutation block matrices $\Psi \in \Pi$ allows for identifiability of the (observed) joint distribution $P_X$.
>
> **4.)** The reviewer is correct that we assume non-symmetric error distributions. This is a stronger assumption than non-Gaussianity, and it indeed excludes Laplace distributions. However, this assumption is necessary for identifiability of the joint distribution, as we explain in detail in our response to reviewer aPgx; also see our response to AWor.
>
> **5.)** We would like to clarify that we already have additional simulations in Appendix F, where we consider a completely different structure with five shared latent variables and fewer domain-specific nodes.
>
> **6.)** Like the reviewer, we believe that it would be nice to compare our methods to prior work. However, previous methods that work on single domains are not applicable: One might be tempted to apply them in each domain separately. However, it would be unclear how to combine the multiple latent causal graphs we get since it is unclear how to determine which variables are shared. Another idea would be to first identify the joint distribution (as we do in Section 3) and then apply the existing algorithms to the joint distribution. The single-domain methods are expected to perform poorly in this setting since they require the classical notion of pure children, whereas we require only "partial" pure children. On the other hand, the algorithm in the proposed reference [1] should be applicable. However, the identifiability results in [1] also require two pure children, which is why we expect it to perform poorly in the considered setups for our simulations. If code is readily available, we will demonstrate this behavior in additional simulations in the Appendix in our camera-ready version.
>
> In our plots, we decided to compare the recovered parameter matrices itself with the correct parameter matrices. If the difference is zero, this, in particular, means that the support of the recovered graph is correctly recovered. On the other hand, recall, precision or f1-score only evaluate the graphical structure. However, we agree with the reviewer that they might give additional insight, and we can easily add these metrics.

---

> > ### Comment · Reviewer_ny5v · 2023-08-14
> >
> > Thanks for the response. I find that the authors have discussed [1] and addressed some problems. However, I think that [2] and [3] have addressed the identification of causal representation, for example, Theorem 4.1 in [3] and Lemma 3.1 in [2], which is suitable for the nonlinear scenario. Compared with existing results, I think the contribution might be limited, so I will keep my score.

---

> > > ### Author Response · Authors · 2023-08-18
> > >
> > > Thanks for pointing us to the specific results in [2] and [3]. They are indeed suitable for a nonlinear setup, but we don't think they address learning a *causal* representation in the way we do.
> > >
> > > In [2], the authors assume a data-generating process in which each observed domain is a function of *shared* content variables $z_C$ and *domain-specific* style variables $z_S^{(i)}$, where both $z_C$ and $z_S^{(i)}$ are latent random vectors. Lemma 3.1 in [2] then shows that the shared latent random vector $z_C$ is block-wise identifiable, i.e., there is an invertible function $h_C$ such that $\hat{z}_C = h_C(z_C)$ for the recovered $\hat{z}_C$. Similar to our case, recovery of the shared latent variables allows for identifiability of the joint distribution. However, the key difference is that we are further interested in the causal relations among the *components* of $z_C$, which are not addressed in [2] but are our central object of study. The authors of [2] only show that the whole vector is identified up to an invertible nonlinear transformation, which does not allow us to draw any conclusions about the causal relations among the individual components of $z_C$. In contrast, we show identifiability of the causal relations (i.e., the graph) between the shared latent variables in Theorem 4.4.
> > >
> > > The work in [3] is a predecessor of [2], and it shows the same difference in the results obtained.

---

> > > > ### Comment · Reviewer_ny5v · 2023-08-18
> > > >
> > > > Thanks for your response! I have found the difference between these works. I will raise the score.

---

### Official Review · Reviewer_B2Nu · 2023-07-04

**Soundness:** 4 excellent
**Presentation:** 4 excellent
**Contribution:** 3 good
**Rating:** 7
**Confidence:** 4

**Summary:**

- The paper considers causal representation learning from unpaired multi-domain data, with latent variables both shared and specific to domains.
- Its key contribution is a new identifiability result for linear causal models with non-Gaussian noise, linear mixing function, and a number of other assumptions.
- In addition, the authors develop a practical representation learning algorithm for this setting and demonstrate it on toy data.

I've read the authors' rebuttal. They have addressed my concerns adequately.

**Strengths:**

- Causal representation learning is an interesting, relevant, and mostly unsolved problem.
- The setting considered here (observational unpaired multi-domain data) is practical and well-motivated from single-cell biology applications.
- The identifiability result is, to the best of my knowledge, novel, and substantially different from existing results.
- As far as I can tell, it is also correct, though due to review overload I have not been able to check the proofs properly.
- The paper is extraordinarily well-written. The authors manage to be precise, yet still provide intuitive explanations.

**Weaknesses:**

- The contribution made here has only one real weakness, and that is the host of strong assumptions underlying the identifiability result: 1D causal variables, linear causal model, no causal effects from shared to domain-specific latents, non-symmetric error distributions, different error distributions for each variable, linear mixing function, full-rank mixing function, observed variables include sufficient "partially pure children", and the list goes on. To put it bluntly, this list makes me wonder if this identifiability result present progress on the road to algorithms that work in practice on interesting real-world datasets.
    - Of course, strong statements such as CRL identifiability require strong inputs, but these need not be in the form of model assumptions, they could also come from the data side. Perhaps it is a bit out of scope for this paper, but I would be curious if the availability of *interventional* data or some other form of auxiliary data would allow the relaxation of some of these model assumptions.
    - While the authors do a good job in providing an intuition for why these assumptions are needed, I would like to know if there are any real-world problems that satisfy them all. This is partially discussed for single-cell data and a few of these assumptions, but could the authors provide a more complete example that ideally satisfies all assumptions?
- There are no experiments to speak of, though I also don't think that all papers need experiments.

**Questions:**

- See above.
- Very minor suggestion: there are a few instances of `\citet{}` that should be a `\citep{}`. It speaks for the quality of the writing that I can't think of any other comments here.

**Limitations:**

- The paper is very clear about the (many) assumptions in the theory. It also openly acknowledges the limitations of the experiments.
- I do not see any particular need for an extensive discussion of societal impacts.

---

> ### Author Rebuttal · Authors · 2023-08-04
>
> We thank the reviewer for their positive comments regarding the quality of our writing, the importance of the studied causal representation learning problem, and the novelty of our result. We hope that the following responses addresses their remaining questions.
>
> # Response to weaknesses
>
> *The contribution made here has only one real weakness, and that is the host of strong assumptions underlying the identifiability result: 1D causal variables, linear causal model, no causal effects from shared to domain-specific latents, non-symmetric error distributions, different error distributions for each variable, linear mixing function, full-rank mixing function, observed variables include sufficient "partially pure children", and the list goes on. To put it bluntly, this list makes me wonder if this identifiability result present progress on the road to algorithms that work in practice on interesting real-world datasets.*
>
> We agree with the reviewer that the linear setting is idealized to give the identifiability problem in an unpaired multi-domain setting a theoretical treatment. However, as we discuss in the response to reviewer aPgx, identifiability in linear setups is often the basis for progress on identifiability results in more general, nonlinear setups. Moreover, the reviewer is correct that we make certain assumptions to obtain the identifiability result. But we want to emphasize that we also strongly focus on discussing their necessity, that is, whether or not they are the weakest assumptions possible. In Appendix C, we discuss the need for non-Gaussianity and sparsity on the mixing matrices. In addition, we again refer to the response to reviewer aPgx for a detailed discussion on the necessity of non-symmetric, pairwise different error distributions.
>
> *Of course, strong statements such as CRL identifiability require strong inputs, but these need not be in the form of model assumptions, they could also come from the data side. Perhaps it is a bit out of scope for this paper, but I would be curious if the availability of interventional data or some other form of auxiliary data would allow the relaxation of some of these model assumptions.*
>
> This is a very interesting point. In this work, we intended to focus purely on the observational case. But for future work, considering interventional data would definitely allow relaxing some conditions in both, recovering the joint distribution and the shared latent graph. Intervention on latent nodes only has an effect on observed domains that are children of that latent node. Thus, we suspect that interventional data allows to identify the shared latent variables and recover the joint distribution. This would imply a relaxation of the distributional assumptions on the errors. Interventional data also allows identification of the shared latent graph without sparsity constraints, as proven in several recent papers on single-domain data (e.g. [1], [2]). Extending this to the multi-domain setup is an interesting problem for future work, and we believe it would allow us to drop the sparsity constraints.
>
> [1] Linear Causal Disentanglement via Interventions
> [2] Interventional Causal Representation Learning
>
> *While the authors do a good job in providing an intuition for why these assumptions are needed, I would like to know if there are any real-world problems that satisfy them all. This is partially discussed for single-cell data and a few of these assumptions, but could the authors provide a more complete example that ideally satisfies all assumptions?*
>
> We thank the reviewer for raising this question. First of all, finding sufficient and necessary conditions for identifiability in causal representation learning is an important problem in itself. Understanding such conditions allows reasoning about whether or not a given data set satisfies these conditions and if causal representation learning is feasible.
>
> While provably certifying assumptions such as the distribution of latent variables and sparsity is challenging, for the data sets motivating our work, i.e., multi-domain data from single-cell biology, it is reasonable to believe that the assumptions are satisfied. This is argued in Remark 4.6, where we also cite a data set. In the remark, we do not discuss the conditions needed for identifiability of the joint distributions. However, assuming that the distributions of the latent variables are pairwise different and non-symmetric, which is for example satisfied almost certainly by randomly chosen probability distributions on the real line and can thus be considered a weak assumption, would be sufficient.
>
> # Questions
>
> *Very minor suggestion: there are a few instances of \citet{} that should be a \citep{}. It speaks for the quality of the writing that I can't think of any other comments here.*
>
> Many thanks for pointing this out; we will change it accordingly.

---

> > ### Comment · Reviewer_B2Nu · 2023-08-10
> >
> > Thanks for the clear response to my review.
> >
> > In particular, you made a good case for studying linear transformations. I'm less convinced that the other assumptions are often met in real-world situations, though.
> >
> > As a concrete example, you pointed again to single-cell data – thanks, I will look into that. Of course that is a field that is relevant in its own right. I am curious though whether there are any other real-world scenarios to which your results apply?
> >
> > > However, assuming that the distributions of the latent variables are pairwise different and non-symmetric, which is for example satisfied almost certainly by randomly chosen probability distributions on the real line and can thus be considered a weak assumption
> >
> > I would like to politely push against this kind of argument. Of course there are many different distributions on the real line. Yet somehow, in reality, many phenomena tend to follow certain few distributions, in particular normal ones (thanks to the central limit theorem, I guess). I would find concrete examples of real-world systems more convincing than arguments based on some measure that is perhaps not very representative of our world.
> >
> > Thanks again for your comments, I look forward to further discussion.

---

> > > ### Author Response · Authors · 2023-08-11
> > >
> > > We thank the reviewer for their comment and agree that Gaussian distributions appear in many scenarios due to the central limit theorem. However, we want to emphasize again that our conditions for identifiability of the joint distribution are sufficient but, at the same time, also necessary. On a higher level, one might interpret this as follows: If one is willing to assume that conceptually different latent factors also follow a different distribution, then identification of these factors is possible, and otherwise not. Said differently, if the assumptions hold, then our method can be applied, and otherwise no method will do well.
> > >
> > > Apart from pairwise different distributions, non-symmetry is then required to fully identify the joint distribution whose dependency structure is determined by the shared latent factors. But if one is not willing to make the additional assumption on non-symmetry (for example, due to many Gaussian real-world scenarios), then it is still possible to identify the shared, conceptually different latent factors.
> > >
> > > This becomes clear by inspecting the proof of Theorem 3.1 and is an important fact that we should add in a remark to the paper. If the error distributions of the latent variables are pairwise different but not necessarily non-symmetric, then the exact joint distribution is in general not identifiable. However, the non-identifiability would only result in sign indeterminacy, that is, the linear effects from the symmetric latents on the domains can be sign-flipped.
> > >
> > > In terms of other real-world scenarios to which our results apply, we want to point out that unpaired multi-domain data appears in many phenomena apart from single-cell biology. For example, images of similar objects are captured in different environments [1], data from multiple domains is common in large biomedical and neuroimaging datasets [2,3,4,5], or stocks are traded in different markets (data can be downloaded from Yahoo Finance). Under the linearity assumption, our results provide conditions under which a shared causal graph is provably identifiable. It then depends on the specific application to reason about whether or not certain assumptions, such as partial pure children, are justifiable. Moreover, as we explained in our previous answer, we consider our results as a basis for progress on identifiability results in nonlinear setups, such as image data.
> > >
> > > [1] Recognition in Terra Incognita \
> > > [2] Multimodal population brain imaging in the UK Biobank prospective epidemiological study \
> > > [3] The WU-Minn Human Connectome Project: an overview \
> > > [4] The Cambridge Centre for Ageing and Neuroscience (Cam-CAN) study protocol: a cross-sectional, lifespan, multidisciplinary examination of healthy cognitive ageing \
> > > [5] Training fMRI Classifiers to Discriminate Cognitive States across Multiple Subjects

---

> > > > ### Comment · Reviewer_B2Nu · 2023-08-16
> > > >
> > > > Thank you for yet another clear response, and for your patience with me.
> > > >
> > > > I really appreciate the point that you also show non-identifiability of certain settings, and that that's a valuable result as well. I also liked the additional examples you gave.
> > > >
> > > > Overall, I am now convinced that the paper makes a valuable contribution to identifiability theory in causal representation learning. It is of a high quality and should be accepted at NeurIPS. I will adapt my score accordingly.

---

### Official Review · Reviewer_aPgx · 2023-07-07

**Soundness:** 3 good
**Presentation:** 3 good
**Contribution:** 3 good
**Rating:** 6
**Confidence:** 3

**Summary:**

This work tackles the problem of learning the latent causal structure from multiple unpaired domains. Under a linear non-Gaussian condition, this work presents the identifiability guarantees for the joint distribution over the domains and the causal structure within the shared latent partition. Synthetic data experiments are presented to validate the theory.

**Strengths:**

1. The problem is well-motivated and timely. Unpaired data are prevalent in the wild, and this work provides a rigorous treatment as the initial step to leverage such data in a principled manner.
2. The paper is nicely written, and the theoretical analysis is clearly articulated with sufficient explanations.
3. The theoretical techniques are clearly explained. Connections and attributions to prior work are appropriately introduced, which aids the assessment of this paper’s technical contribution.

**Weaknesses:**

1. The linear assumptions: practical multi-domain (modal) data-generating processes are often highly nonlinear, e.g., images and text. The applicability of the linear assumption may not be as appealing.
2. The heterogeneous noise distributions: pairwise distinct exogenous distributions appear a strong assumption to me and can potentially oversimplify the technical challenge. I would be interested in learning about the necessity of such an assumption.

**Questions:**

I would like to learn about the authors' response to the weaknesses listed above, which may give me a clearer perspective on the paper's contribution.

**Limitations:**

Please see the weakness section.

---

> ### Author Rebuttal · Authors · 2023-08-02
>
> We thank the reviewer for their positive review of our work and their supportive comments on the motivation of the problem and the presentation of the paper. We hope that the following responses addresses their remaining questions.
>
> # Response to the weaknesses
>
> ### Linearity
> We agree that the linear setting is somewhat idealized to give the identifiability problem in an unpaired multi-domain setting a theoretical treatment. In this paper, we focus on understanding the problem in the linear case, which we consider the basis for any subsequent study of nonlinear cases. For example, recent advances in nonlinear ICA allow identifiability up to linear transformations, see e.g. [1] and [2]. Identifiability of a *causal* representation might then be obtained from identifiability results for the linear model.
>
> This is supported by examples in recent work where identifiability in the linear case led to significant progress to obtain identifiability in more general models. For example, the authors of [3] consider identifiability under interventional data in a nonlinear setup. They show identifiability by first proving identifiability up to linear transformations and then applying an identifiability result for linear setups [4].
>
> To conclude, we consider it as a very interesting follow-up work to study a nonlinear multi-domain setting, where we can "combine" identifiability of nonlinear ICA with our result on linear identifiability in multi-domains.
>
> [1] Variational Autoencoders and Nonlinear ICA: A Unifying Framework \
> [2] On Linear Identifiability of Learned Representations \
> [3] Learning Linear Causal Representations from Interventions under General Nonlinear Mixing \
> [4] Linear Causal Disentanglement via Interventions
>
> ### Noise distribution assumption
> Many thanks for raising the question of the necessity of pairwise different error distributions. Our paper proposes a constructive way to recover the joint distribution in a multi-domain causal representation model (Definition 2.3) from the marginal distributions. As pointed out, this relies on the assumption of pairwise distinct error distributions. Inspired by the reviewer's comment, we discuss the necessity in detail in the following.
>
> **Recap of identifiability**\
> In general, given two potentially different multi-domain graphs $G_m$ ans $\widetilde{G}_m$ and any representations $(B,P)$ and $(\widetilde{B}, \widetilde{P})$, identifiability of the joint distributions in multi-domain causal representation models means the following:
>
> $B_{V_e,H}$#$P=\widetilde{B}_{V_e,H}$#$\widetilde{P}$ for all $e \in [m]$ $\implies$ $B$#$P=\widetilde{B}$#$\widetilde{P}$,
>
> where $M$#$P$ denotes the push-forward measure under the linear map $M$. The left-hand side means that the marginal distributions in each domain are equal, while the right-hand side means that the joint distributions are equal. Note that the right-hand side holds if and only if there is a signed permutation matrix $\Psi$ (not necessarily with block structure) such that $B=\widetilde{B}\Psi$ and $P = \Psi^{\top}$#$\widetilde{P}$. This is implied by the identifiability of the usual, one-domain linear ICA.
>
> Theorem 3.1 shows that assumptions (C1) and (C2) imply identifiability of the joint distribution. In particular, we show that they imply identifiability in a stronger sense, namely, that it follows from the left-hand side that $B=\widetilde{B}\Psi$ and $P = \Psi^{\top}$#$\widetilde{P}$ for a signed permutation *block* matrix $\Psi \in \Pi$.  Identification of the representation $(B,P)$ up to the equivalence class of signed permutation *block* matrices implies identifiability of the joint distribution, and it also allows for identifying the shared causal graph in a second step.
>
> Now, it is possible to discuss the necessity of pairwise distinct error distributions in two contexts: Is the assumption necessary for identifiability of the representation $(B,P)$ up to signed permutation block matrices? Is it necessary for identifiability of the joint distribution?
>
> **Role of distributional assumptions**\
> Assume that $P_i = P_j$ or $P_i = -P_j$. Then four cases are possible: \
> (i) $i,j \in \mathcal{L}$ \
> (ii) $i \in \mathcal{L}, j \in I_e$ \
> (iii) $i \in I_e, j \in I_f$ for $i \neq j$ \
> (iv) $i,j \in I_e$
>
> In cases (i) - (iii), identification of the representation $(B,P)$ up to a signed permutation block matrix is not possible. Moreover, in these three cases, it is also impossible to identify the joint distribution at all.
>
> In **cases (i)-(ii)**, this can be seen by construction matrices $B$ and $\widetilde{B}$ that can not be transformed into each other by any signed permutation matrix (without block structure), which implies that $B$#$P \neq \widetilde{B}$#$\widetilde{P}$, but the marginal distributions are equivalent.
>
> In **case (iii)**, it can be seen by constructing a counterexample where two joint distributions are supported on different dimensional subspaces, but the marginal distributions are again equivalent.
>
> ***To summarize, pairwise different error distributions are necessary in cases (i) - (iii) to identify the joint distribution.*** We are more than happy to extend Appendix C and include this discussion and the counterexamples in our camera-ready version.
>
> On the other hand, **case (iv)** would still allow identification of the representation $(B,P)$ up to a signed permutation block matrix and, therefore, also identification of the joint distribution. For the sake of a simplified presentation, we were not stating this explicitly in the paper, even though it becomes clear by checking the proof of Theorem 3.1. We will add a remark in the main body of the camera-ready version to clarify.

---

> > ### Comment · Reviewer_aPgx · 2023-08-13
> >
> > Many thanks for the thoughtful response. While I still think certain assumptions are a bit unrealistic (e.g., linearity), I gained a better understanding of the pairwise difference assumption, thanks to the response. I will keep my rating as it is.

---

### Decision · Program_Chairs · 2023-09-21

**Decision:**

Accept (spotlight)

**Comment:**

This research addresses the challenge of discerning the latent causal relationships across multiple unrelated domains. In the context of a linear non-Gaussian scenario, this study provides assurances regarding the identifiability of the joint distribution spanning the domains and the causal structure within the common latent segment. To substantiate these theoretical findings, the paper includes synthetic data experiments.

The paper addresses a timely and well-motivated problem concerning the utilization of unpaired data, which is abundant in real-world scenarios. It offers a rigorous approach to leveraging such data in a principled manner, making it an important initial step. Hopefully more general results in the nonlinear setting can arise in the near future. All reviewers unanimously recommend acceptance of this work. The paper is well-written, and its theoretical analysis is presented clearly and with ample explanations. The explanations of the theoretical techniques are also well-detailed. I thus recommend acceptance of this paper.